# Ophiuchus: Scalable Modeling of Protein Structures through Hierarchical Coarse-graining SO(3)-Equivariant Autoencoders

## Abstract

Three-dimensional native states of natural proteins display recurring and hierarchical patterns. Yet, traditional graph-based modeling of protein structures is often limited to operate within a single fine-grained resolution, and lacks hourglass neural architectures to learn those high-level building blocks. We narrow this gap by introducing Ophiuchus, an SO(3)-equivariant coarse-graining model that efficiently operates on all-atom protein structures. Our model departs from current approaches that employ graph modeling, instead focusing on local convolutional coarsening to model sequence-motif interactions with efficient time complexity in protein length. We measure the reconstruction capabilities of Ophiuchus across different compression rates, and compare it to existing models. We examine the learned latent space and demonstrate its utility through conformational interpolation. Finally, we leverage denoising diffusion probabilistic models (DDPM) in the latent space to efficiently sample protein structures. Our experiments demonstrate Ophiuchus to be a scalable basis for efficient protein modeling and generation.

## 1 Introduction

Proteins form the basis of all biological processes and understanding them is critical to biological discovery, medical research and drug development. Their three-dimensional structures often display modular organization across multiple scales, making them promising candidates for modeling in motif-based design spaces [Bystroff & Baker (1998); Mackenzie & Grigoryan (2017); Swanson et al. (2022)]. Harnessing these coarser, lower-frequency building blocks is of great relevance to the investigation of the mechanisms behind protein evolution, folding and dynamics [Mackenzie et al. (2016)], and may be instrumental in enabling more efficient computation on protein structural data through coarse and latent variable modeling [Kmiecik et al. (2016); Ramaswamy et al. (2021)].

Recent developments in deep learning architectures applied to protein sequences and structures demonstrate the remarkable capabilities of neural models in the domain of protein modeling and design [Jumper et al. (2021); Baek et al. (2021b); Ingraham et al. (2022); Watson et al. (2022)]. Still, current state-of-the-art architectures lack the structure and mechanisms to directly learn and operate on modular protein blocks.

To fill this gap, we introduce Ophiuchus, a deep SO(3)-equivariant model that captures joint encodings of sequence-structure motifs of all-atom protein structures. Our model is a novel autoencoder that uses one-dimensional sequence convolutions on geometric features to learn coarsened representations of proteins. Ophiuchus outperforms existing SO(3)-equivariant autoencoders [Fu et al. (2023)] on the protein reconstruction task. We present extensive ablations of model performance across different autoencoder layouts and compression settings. We demonstrate that our model learns a robust and structured representation of protein structures by learning a denoising diffusion probabilistic model (DDPM) [Ho et al. (2020)] in the latent space. We find Ophiuchus to enable significantly faster sampling of protein structures, as compared to existing diffusion models [Wu et al. (2022a); Yim et al. (2023); Watson et al. (2023)], while producing unconditional samples of comparable quality and diversity.

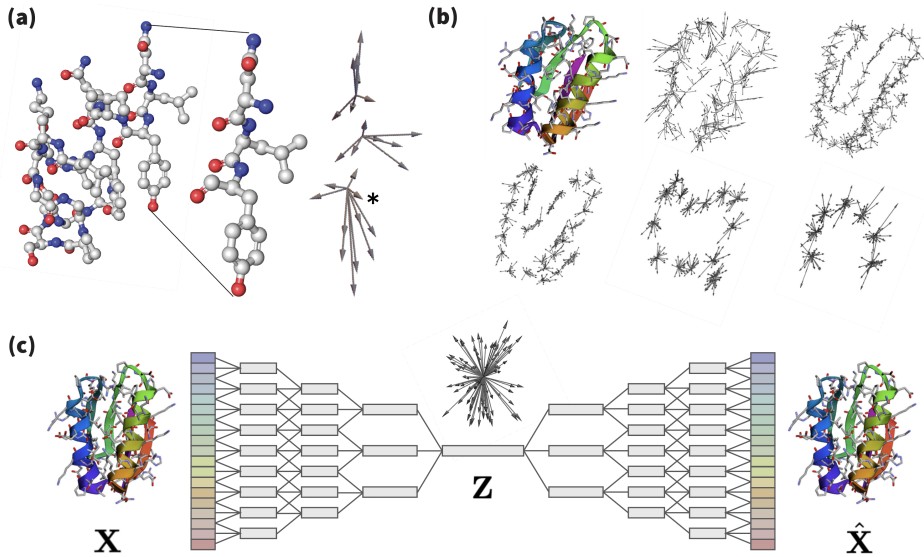

Figure 1: **Coarsening a Three-Dimensional Sequence**. **(a)** Each residue is represented with its $C_\alpha$ atom position and geometric features that encode its label and the positions of its other atoms. **(b)** Our proposed model uses roto-translation equivariant convolutions to coarsen these positions and geometric features. **(c)** We train deep autoencoders to reconstruct all-atom protein structures directly in three-dimensions.

Our main contributions are summarized as follows:

- **Novel Autoencoder**: We introduce a novel SO(3)-equivariant autoencoder for protein sequence and all-atom structure representation. We propose novel learning algorithms for coarsening and refining protein representations, leveraging irreducible representations of SO(3) to efficiently model geometric information. We demonstrate the power of our latent space through unsupervised clustering and latent interpolation.

- **Extensive Ablation**: We offer an in-depth examination of our architecture through extensive ablation across different protein lengths, coarsening resolutions and model sizes. We study the trade-off of producing a coarsened representation of a protein at different resolutions and the recoverability of its sequence and structure.

- **Latent Diffusion**: We explore a novel generative approach to proteins by performing latent diffusion on geometric feature representations. We train diffusion models for multiple resolutions, and provide diverse benchmarks to assess sample quality. To the best of our knowledge, this is the first generative model to directly produce all-atom structures of proteins.

## 2 BACKGROUND AND RELATED WORK

### 2.1 MODULARITY AND HIERARCHY IN PROTEINS

Protein sequences and structures display significant degrees of modularity. [Vallat et al. (2015)] introduces a library of common super-secondary structural motifs (Smotifs), while [Mackenzie et al. (2016)] shows protein structural space to be efficiently describable by small tertiary alphabets (TERMs). Motif-based methods have been successfully used in protein folding and design [Bystroff & Baker (1998); Li et al. (2022)]. Inspired by this hierarchical nature of proteins, our proposed model learns coarse-grained representations of protein structures.

## 2.2 Symmetries in Neural Architecture for Biomolecules

Learning algorithms greatly benefit from proactively exploiting symmetry structures present in their data domain [Bronstein et al. (2021); Smidt (2021)]. In this work, we investigate three relevant symmetries for the domain of protein structures:

**Euclidean Equivariance of Coordinates and Feature Representations**. Neural models equipped with roto-translational (Euclidean) invariance or equivariance have been shown to outperform competitors in molecular and point cloud tasks [Townshend et al. (2022); Miller et al. (2020); Deng et al. (2021)]. Similar results have been extensively reported across different structural tasks of protein modeling [Liu et al. (2022); Jing et al. (2021)]. Our proposed model takes advantage of Euclidean equivariance both in processing of coordinates and in its internal feature representations, which are composed of scalars and higher order geometric tensors [Thomas et al. (2018); Weiler et al. (2018)].

**Translation Equivariance of Sequence**. One-dimensional Convolutional Neural Networks (CNNs) have been demonstrated to successfully model protein sequences across a variety of tasks [Karydis (2017); Hou et al. (2018); Lee et al. (2019); Yang et al. (2022)]. These models capture sequence-motif representations that are equivariant to translation of the sequence. However, sequential convolution is less common in architectures for protein structures, which are often cast as Graph Neural Networks (GNNs) [Zhang et al. (2021)]. Notably, [Fan et al. (2022)] proposes a CNN network to model the regularity of one-dimensional sequences along with three-dimensional structures, but they restrict their layout to coarsening. In this work, we further integrate geometry into sequence by directly using three-dimensional vector feature representations and transformations in 1D convolutions. We use this CNN to investigate an autoencoding approach to protein structures.

**Permutation Invariances of Atomic Order**. In order to capture the permutable ordering of atoms, neural models of molecules are often implemented with permutation-invariant GNNs [Wieder et al. (2020)]. Nevertheless, protein structures are sequentially ordered, and most standard side-chain heavy atoms are readily orderable, with exception of four residues [Jumper et al. (2021)]. We use this fact to design an efficient approach to directly model all-atom protein structures, introducing a method to parse atomic positions in parallel channels as roto-translational equivariant feature representations.

## 2.3 Unsupervised Learning of Proteins

Unsupervised techniques for capturing protein sequence and structure have witnessed remarkable advancements in recent years [Lin et al. (2023); Elnaggar et al. (2023); Zhang et al. (2022)]. Amongst unsupervised methods, autoencoder models learn to produce efficient low-dimensional representations using an informational bottleneck. These models have been successfully deployed to diverse protein tasks of modeling and sampling [Eguchi et al. (2020); Lin et al. (2021); Wang et al. (2022); Mansoor et al. (2023); Visani et al. (2023)], and have received renewed attention for enabling the learning of coarse representations of molecules [Wang & Gómez-Bombarelli (2019); Yang & Gómez-Bombarelli (2023); Winter et al. (2021); Wehmeyer & Noé (2018); Ramaswamy et al. (2021)]. However, existing three-dimensional autoencoders for proteins do not have the structure or mechanisms to explore the extent to which coarsening is possible in proteins. In this work, we fill this gap with extensive experiments on an autoencoder for deep protein representation coarsening.

## 2.4 Denoising Diffusion for Proteins

Denoising Diffusion Probabilistic Models (DDPM) [Sohl-Dickstein et al. (2015); Ho et al. (2020)] have found widespread adoption through diverse architectures for generative sampling of protein structures. Chroma [Ingraham et al. (2022)] trains random graph message passing through roto-translational invariant features, while RFDiffusion [Watson et al. (2022)] fine-tunes pretrained folding model RoseTTAFold [Baek et al. (2021a)] to denoising, employing SE(3)-equivariant transformers in structural decoding [Fuchs et al. (2020)]. [Yim et al. (2023); Anand & Achim (2022)] generalize denoising diffusion to frames of reference, employing Invariant Point Attention [Jumper et al. (2021)] to model three dimensions, while FoldingDiff [Wu et al. (2022a)] explores denoising in angular space. More recently, [Fu et al. (2023)] proposes a latent diffusion model on coarsened representations learned through Equivariant Graph Neural Networks (EGNN) [Satorras et al.

(2022)]. In contrast, our model uses roto-translation equivariant features to produce increasingly richer structural representations from autoencoded sequence and coordinates. We propose a novel latent diffusion model that samples directly in this space for generating protein structures.

## 3 THE OPHIUCHUS ARCHITECTURE

We represent a protein as a sequence of $N$ residues each with an anchor position $\mathbf{P} \in \mathbb{R}^{1 \times 3}$ and a tensor of irreducible representations of SO(3) $\mathbf{V}^{0:l_{\max}}$, where $\mathbf{V}^l \in \mathbb{R}^{d \times (2l+1)}$ and degree $l \in [0, l_{\max}]$. A residue state is defined as $(\mathbf{P}, \mathbf{V}^{0:l_{\max}})$. These representations are directly produced from sequence labels and all-atom positions, as we describe in the **Atom Encoder/Decoder** sections. To capture the diverse interactions within a protein, we propose three main components. **Self-Interaction** learns geometric representations of each residue independently, modeling local interactions of atoms within a single residue. **Sequence Convolution** simultaneously updates sequential segments of residues, modeling inter-residue interactions between sequence neighbors. Finally, **Spatial Convolution** employs message-passing of geometric features to model interactions of residues that are nearby in 3D space. We compose these three modules to build an hourglass architecture.

### 3.1 ALL-ATOM ATOM ENCODER AND DECODER

Given a particular $i$-th residue, let $\mathbf{R}_i \in \mathcal{R}$ denote its residue label, $\mathbf{P}_i^\alpha \in \mathbb{R}^{1 \times 3}$ denote the global position of its alpha carbon ($\mathbf{C}_\alpha$), and $\mathbf{P}_i^* \in \mathbb{R}^{n \times 3}$ the position of all $n$ other atoms relative to $\mathbf{C}_\alpha$. We produce initial residue representations $(\mathbf{P}, \mathbf{V}^{0:l_{\max}})_i$ by setting anchors $\mathbf{P}_i = \mathbf{P}_i^\alpha$, scalars $\mathbf{V}_i^{l=0} = \text{Embed}(\mathbf{R}_i)$ , and geometric vectors $\mathbf{V}_i^{l>0}$ to explicitly encode relative atomic positions $\mathbf{P}_i^*$.

In particular, provided the residue label $\mathbf{R}$, the heavy atoms of most standard protein residues are readily put in a canonical order, enabling direct treatment of atom positions as a stack of signals on SO(3): $\mathbf{V}^{l=1} = \mathbf{P}^*$. However, some of the standard residues present two-permutations within their atom configurations, in which pairs of atoms have ordering indices $(v, u)$ that may be exchanged (Appendix A.1). To handle these cases, we instead use geometric vectors to encode the center $\mathbf{V}_{\text{center}}^{l=1} = \frac{1}{2}(\mathbf{P}_v^* + \mathbf{P}_u^*)$ and the unsigned difference $\mathbf{V}_{\text{diff}}^{l=2} = Y_2(\frac{1}{2}(\mathbf{P}_v^* - \mathbf{P}_u^*))$ between the positions of the pair, where $Y_2$ is a spherical harmonics projector of degree $l = 2$. This signal is invariant to corresponding atomic two-flips, while still directly carrying information about positioning and angularity. To invert this encoding, we invert a signal of degree $l = 2$ into two arbitrarily ordered vectors of degrees $l = 1$. Please refer to Appendix A.2 for further details.

| **Algorithm 1:** ALL-ATOM ENCODING | **Algorithm 2:** ALL-ATOM DECODING |
|---|---|
| **Input:** $\mathbf{C}_\alpha$ Position $\mathbf{P}^\alpha \in \mathbb{R}^{1 \times 3}$ 
 **Input:** All-Atom Relative Positions 
 $\quad\quad \mathbf{P}^* \in \mathbb{R}^{n \times 3}$ 
 **Input:** Residue Label $\mathbf{R} \in \mathcal{R}$ 
 **Output:** Latent Representation $(\mathbf{P}, \mathbf{V}^{l=0:2})$ 
 $\mathbf{P} \leftarrow \mathbf{P}^\alpha$ 
 $\mathbf{V}^{l=0} \leftarrow \text{Embed}(\mathbf{R})$ 
 $\mathbf{V}_{\text{ordered}}^{l=1} \leftarrow \text{GetOrderablePositions}(\mathbf{R}, \mathbf{P}^*)$ 
 $\mathbf{P}_v^*, \mathbf{P}_u^* \leftarrow \text{GetUnorderablePositionPairs}(\mathbf{R}, \mathbf{P}^*)$ 
 $\mathbf{V}_{\text{center}}^{l=1} \leftarrow \frac{1}{2}(\mathbf{P}_v^* + \mathbf{P}_u^*)$ 
 $\mathbf{V}_{\text{diff}}^{l=2} \leftarrow Y^2\left(\frac{1}{2}(\mathbf{P}_v^* - \mathbf{P}_u^*)\right)$ 
 $\mathbf{V}^{l=0:2} \leftarrow \mathbf{V}_{\text{ordered}}^{l=1} \oplus \mathbf{V}_{\text{center}}^{l=1} \oplus \mathbf{V}_{\text{diff}}^{l=2}$ 
 **return** $(\mathbf{P}, \mathbf{V}^{l=0:2})$ | **Input:** Latent Representation $(\mathbf{P}, \mathbf{V}^{l=0:2})$ 
 **Output:** $\mathbf{C}_\alpha$ Position $\mathbf{P}^\alpha \in \mathbb{R}^{1 \times 3}$ 
 **Output:** All-Atom Relative Positions 
 $\quad\quad \mathbf{P}^* \in \mathbb{R}^{|\mathcal{R}| \times n \times 3}$ 
 **Output:** Residue Label Logits $\boldsymbol{\ell} \in \mathbb{R}^{|\mathcal{R}|}$ 
 $\mathbf{P}^\alpha \leftarrow \mathbf{P}$ 
 $\boldsymbol{\ell} \leftarrow \text{LogSoftmax}(\text{Linear}(\mathbf{V}^{l=0}))$ 
 $\hat{\mathbf{V}}_{\text{ordered}}^{l=1} \leftarrow \text{Linear}(\mathbf{V}^{l=1})$ 
 $\hat{\mathbf{V}}_{\text{center}}^{l=1}, \hat{\mathbf{V}}_{\text{diff}}^{l=2} \leftarrow \text{Linear}(\mathbf{V}^{l=0:2})$ 
 $\Delta\mathbf{P}_{v,u} \leftarrow \text{Eigendecompose}(\hat{\mathbf{V}}_{\text{diff}}^{l=2})$ 
 $\mathbf{P}_v^*, \mathbf{P}_u^* = \hat{\mathbf{V}}_{\text{center}}^{l=1} + \Delta\mathbf{P}_{v,u}, \hat{\mathbf{V}}_{\text{center}}^{l=1} - \Delta\mathbf{P}_{v,u}$ 
 $\mathbf{P}^* \leftarrow \mathbf{P}^* \oplus \mathbf{P}_v^* \oplus \mathbf{P}_u^*$ 
 **return** $(\mathbf{P}^\alpha, \mathbf{P}^*, \boldsymbol{\ell})$ |

This processing makes Ophiuchus strictly blind to ordering flips of permutable atoms, while still enabling it to operate directly on all atoms in an efficient, stacked representation. In Appendix A.1, we illustrate how this approach correctly handles the geometry of side-chain atoms.

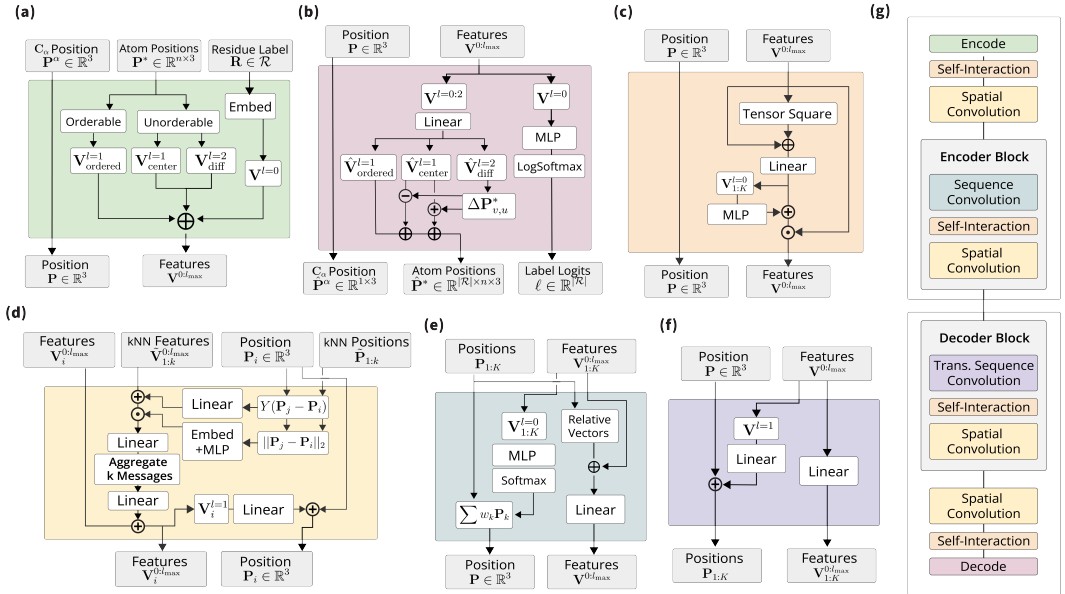

Figure 2: **Building Blocks of Ophiuchus**: **(a) Atom Encoder** and **(b) Atom Decoder** enable the model to directly take and produce atomic coordinates. **(c) Self-Interaction** updates representations internally, across different vector orders $l$. **(d) Spatial Convolution** interacts spatial neighbors. **(e) Sequence Convolution** and **(f) Transpose Sequence Convolution** communicate sequence neighbors and produce coarser and finer representations, respectively. **(g) Hourglass model**: we compose those modules to build encoder and decoder models, stacking them into an autoencoder.

## 3.2 SELF-INTERACTION

Our Self-Interaction is designed to model the internal interactions of atoms within each residue. This transformation updates the feature vectors $\mathbf{V}_i^{0:l_{\max}}$ centered at the same residue $i$. Importantly, it blends feature vectors $\mathbf{V}^l$ of varying degrees $l$ by employing tensor products of the features with themselves. We offer two implementations of these tensor products to cater to different computational needs. Our Self-Interaction module draws inspiration from MACE [Batatia et al. (2022)]. For a comprehensive explanation, please refer to Appendix A.3.

---

**Algorithm 3:** Self-Interaction

**Input:** Latent Representation $(\mathbf{P}, \mathbf{V}^{0:l_{\max}})$

$\mathbf{V}^{0:l_{\max}} \leftarrow \mathbf{V}^{0:l_{\max}} \oplus \left(\mathbf{V}^{0:l_{\max}}\right)^{\otimes 2}$      ▷ Tensor Square and Concatenate

$\mathbf{V}^{0:l_{\max}} \leftarrow \text{Linear}(\mathbf{V}^{0:l_{\max}})$          ▷ Update features

$\mathbf{V}^{0:l_{\max}} \leftarrow \text{MLP}(\mathbf{V}^{l=0}) \cdot \mathbf{V}^{0:l_{\max}}$     ▷ Gate Activation Function

**return** $(\mathbf{P}, \mathbf{V}^{0:l_{\max}})$

---

## 3.3 SEQUENCE CONVOLUTION

To take advantage of the sequential nature of proteins, we propose a one-dimensional, roto-translational equivariant convolutional layer for acting on geometric features and positions of sequence neighbors. Given a kernel window size $K$ and stride $S$, we concatenate representations $\mathbf{V}_{i-\frac{K}{2}:i+\frac{K}{2}}^{0:l_{\max}}$ with the same $l$ value. Additionally, we include normalized relative vectors between anchoring positions $\mathbf{P}_{i-\frac{K}{2}:i+\frac{K}{2}}$. Following conventional CNN architectures, this concatenated representation undergoes a linear transformation. The scalars in the resulting representation are then converted into weights, which are used to combine window coordinates into a new coordinate. To ensure translational equivariance, these weights are constrained to sum to one.

---

**Algorithm 4:** Sequence Convolution

---

**Input:** Window of Latent Representations $(\mathbf{P}, \mathbf{V}^{0:l_{\max}})_{1:K}$

$w_{1:K} \leftarrow \text{Softmax}\left(\text{MLP}(\mathbf{V}^0_{1:K})\right)$         ▷ Such that $\sum_k w_k = 1$

$\mathbf{P} \leftarrow \sum_{k=1}^{K} w_k \mathbf{P}_k$         ▷ Coarsen coordinates

$\tilde{\mathbf{V}} \leftarrow \bigoplus_K \mathbf{V}^{0:l_{\max}}_{1:K}$         ▷ Stack features

$\tilde{\mathbf{P}} \leftarrow \bigoplus_{i=1,j=1}^{K,K} Y\left(\mathbf{P}_i - \mathbf{P}_j\right)$         ▷ Stack relative vectors

$\mathbf{V}^{0:l_{\max}} \leftarrow \text{Linear}\left(\tilde{\mathbf{V}} \oplus \tilde{\mathbf{P}}\right)$         ▷ Coarsen features and vectors

**return** $(\mathbf{P}, \mathbf{V}^{0:l_{\max}})$

---

When $S > 1$, sequence convolutions reduce the dimensionality along the sequence axis, yielding mixed representations and coarse coordinates. To reverse this procedure, we introduce a transpose convolution algorithm that uses its $\mathbf{V}^{l=1}$ features to spawn coordinates. For further details, please refer to Appendix A.6.

## 3.4 SPATIAL CONVOLUTION

To capture interactions of residues that are close in three-dimensional space, we introduce the Spatial Convolution. This operation updates representations and positions through message passing within k-nearest spatial neighbors. Message representations incorporate SO(3) signals from the vector difference between neighbor coordinates, and we aggregate messages with a permutation-invariant means. After aggregation, we linearly transform the vector representations into a an update for the coordinates.

---

**Algorithm 5:** Spatial Convolution

---

**Input:** Latent Representations $(\mathbf{P}, \mathbf{V}^{0:l_{\max}})_{1:N}$
**Input:** Output Node Index $i$

$(\tilde{\mathbf{P}}, \tilde{\mathbf{V}}^{0:l_{\max}})_{1:k} \leftarrow k\text{-Nearest-Neighbors}(\mathbf{P}_i, \mathbf{P}_{1:N})$

$R_{1:k}, \; \phi_{1:k} \leftarrow \text{Embed}(||\tilde{\mathbf{P}}_{1:k} - \mathbf{P}_i||_2), \; Y(\tilde{\mathbf{P}}_{1:k} - \mathbf{P}_i)$         ▷ Edge Features

$\tilde{\mathbf{V}}^{0:l_{\max}}_{1:k} \leftarrow \text{MLP}(R_k) \cdot \left(\text{Linear}(\tilde{\mathbf{V}}^{0:l_{\max}}_{1:k}) + \text{Linear}(\phi_{1:k})\right)$         ▷ Prepare messages

$\mathbf{V}^{0:l_{\max}} \leftarrow \text{Linear}\left(\mathbf{V}^{0:l_{\max}}_i + \frac{1}{k}\left(\sum_k \tilde{\mathbf{V}}^{0:l_{\max}}_k\right)\right)$         ▷ Aggregate and update

$\mathbf{P} \leftarrow \mathbf{P}_i + \text{Linear}\left(\mathbf{V}^{l=1}\right)$         ▷ Update positions

**return** $(\mathbf{P}, \mathbf{V}^{0:l_{\max}})$

---

## 3.5 DEEP COARSENING AUTOENCODER

We compose Space Convolution, Self-Interaction and Sequence Convolution modules to define a coarsening/refining block. The block mixes representations across all relevant axes of the domain while producing coarsened, downsampled positions and mixed embeddings. The reverse result – finer positions and decoupled embeddings – is achieved by changing the standard Sequence Convolution to its transpose counterpart. When employing sequence convolutions of stride $S > 1$, we increase the dimensionality of the feature representation according to a rescaling factor hyperparameter $\rho$. We stack $L$ coarsening blocks to build a deep neural encoder $\mathcal{E}$ (Alg.7), and symetrically $L$ refining blocks to build a decoder $\mathcal{D}$ (Alg. 8).

## 3.6 AUTOENCODER RECONSTRUCTION LOSSES

We use a number of reconstruction losses to ensure good quality of produced proteins.

**Vector Map Loss**. We train the model by directly comparing internal three-dimensional vector difference maps. Let $V(\mathbf{P})$ denote the internal vector map between all atoms $\mathbf{P}$ in our data, that is, $V(\mathbf{P})^{i,j} = (\mathbf{P}^i - \mathbf{P}^j) \in \mathbb{R}^3$. We define the vector map loss as $\mathcal{L}_{\text{VectorMap}} = \text{HuberLoss}(V(\mathbf{P}), V(\hat{\mathbf{P}}))$ [Huber (1992)]. When computing this loss, an additional stage is employed for processing permutation symmetry breaks. More details can be found in Appendix B.1.

**Residue Label Cross Entropy Loss**. We train the model to predict logits $\ell$ over alphabet $\mathcal{R}$ for each residue. We use the cross entropy between predicted logits and ground labels: $\mathcal{L}_{\text{CrossEntropy}} = \text{CrossEntropy}(\ell, \mathbf{R})$

**Chemistry Losses**. We incorporate $L_2$ norm-based losses for comparing bonds, angles and dihedrals between prediction and ground truth. For non-bonded atoms, a clash loss is evaluated using standard Van der Waals atomic radii (Appendix B.2)

Please refer to Appendix B for further details on the loss.

### 3.7 LATENT DIFFUSION

We train an SO(3)-equivariant DDPM [Ho et al. (2020); Sohl-Dickstein et al. (2015)] on the latent space of our autoencoder. We pre-train an autoencoder and transform each protein $\mathbf{X}$ from the dataset into a geometric tensor of irreducible representations of SO(3): $\mathbf{Z} = \mathcal{E}(\mathbf{X})$. We attach a diffusion process of $T$ steps on the latent variables, making $\mathbf{Z}_0 = \mathbf{Z}$ and $\mathbf{Z}_T = \mathcal{N}(0, 1)$. We follow the parameterization described in [Salimans & Ho (2022)], and train a denoising model to reconstruct the original data $\mathbf{Z}_0$ from its noised version $\mathbf{Z}_t$:

$$\mathcal{L}_{\text{diffusion}} = \mathbb{E}_{\epsilon, t}\left[ w(\lambda_t) || \hat{\mathbf{Z}}_0(\mathbf{Z}_t) - \mathbf{Z}_0 ||_2^2 \right]$$

In order to ensure that bottleneck representations $\mathbf{Z}_0$ are well-behaved for generation purposes, we regularize the latent space of our autoencoder (Appendix B.3). We build a denoising network with $L_D$ layers of Self-Interaction. Please refer to Appendix F for further details.

### 3.8 IMPLEMENTATION

We train all models on single-GPU A6000 and GTX-1080 Ti machines. We implement Ophiuchus in Jax using the python libraries e3nn-jax [Geiger & Smidt (2022)] and Haiku [Hennigan et al. (2020)].

## 4 METHODS AND RESULTS

### 4.1 AUTOENCODER ARCHITECTURE COMPARISON

We compare Ophiuchus to the architecture proposed in [Fu et al. (2023)], which uses the EGNN-based architecture [Satorras et al. (2022)] for autoencoding protein backbones. To the best of our knowledge, this is the only other model that attempts protein reconstruction in three-dimensions with roto-translation equivariant networks. For demonstration purposes, we curate a small dataset of protein $\mathbf{C}_\alpha$-backbones from the PDB with lengths between 16 and 64 and maximum resolution of up to 1.5 Å, resulting in 1267 proteins. We split the data into train, validation and test sets with ratio [0.8, 0.1, 0.1]. In table 1, we report the test performance at best validation step, while avoiding over-fitting during training.

Table 1: **Reconstruction from single feature bottleneck**

| Model | Downsampling Factor | Channels/Layer | # Params [1e6] $\downarrow$ | $C\alpha$-RMSD (Å) $\downarrow$ | Residue Acc. (%) $\uparrow$ |
|---|---|---|---|---|---|
| EGNN | 2 | [32, 32] | 0.68 | 1.01 | 88 |
| EGNN | 4 | [32, 32, 48] | 1.54 | 1.12 | 80 |
| EGNN | 8 | [32, 32, 48, 72] | 3.30 | 2.06 | 73 |
| EGNN | 16 | [32, 32, 48, 72, 108] | 6.99 | 11.4 | 25 |
| Ophiuchus | 2 | [5, 7] | 0.018 | 0.11 | 98 |
| Ophiuchus | 4 | [5, 7, 10] | 0.026 | 0.14 | 97 |
| Ophiuchus | 8 | [5, 7, 10, 15] | 0.049 | 0.36 | 79 |
| Ophiuchus | 16 | [5, 7, 10, 15, 22] | 0.068 | 0.43 | 37 |

We find that Ophiuchus vastly outperforms the EGNN-based architecture. Ophiuchus recovers the protein sequence and $\mathbf{C}_\alpha$ backbone with significantly better, while using orders of magnitude less parameters. Refer to Appendix C for further details.

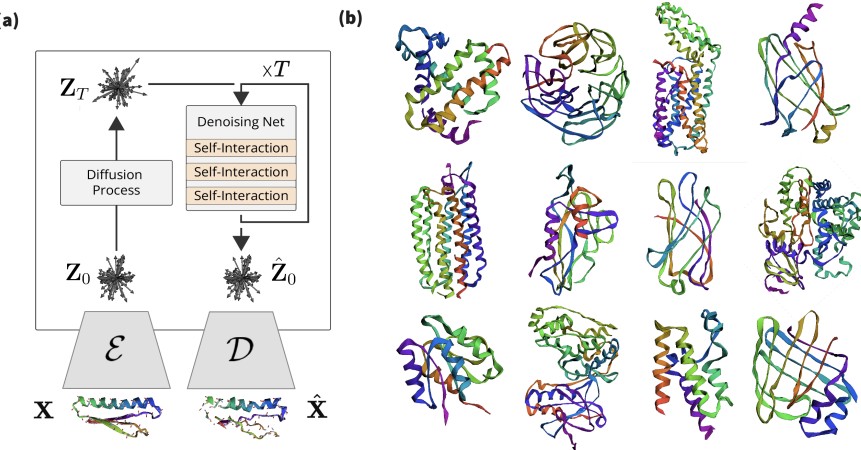

Figure 3: **Protein Latent Diffusion**. **(a)** We attach a diffusion process to Ophiuchus representations and learn a denoising network to sample embeddings of SO(3) that decode to protein structures. **(b)** Random samples from 485-length backbone model.

## 4.2 ARCHITECTURE ABLATION

To investigate the effects of different architecture layouts and coarsening rates, we train different instantiations of Ophiuchus to coarsen representations of contiguous 160-sized protein fragments from the Protein Databank (PDB) [Berman et al. (2000)]. We filter out entries tagged with resolution higher than 2.5 Å, and total sequence lengths larger than 512. We also ensure proteins in the dataset have same chirality. For ablations, the sequence convolution uses kernel size of 5 and stride 3, channel rescale factor per layer is one of {1.5, 1.7, 2.0}, and the number of downsampling layers ranges in 3-5. The initial residue representation uses 16 channels, where each channel is composed of one scalar ($l = 0$) and one 3D vector ($l = 1$). All experiments were repeated 3 times with different initialization seeds and data shuffles.

Table 2: **Ophiuchus Recovery from Compressed Representations - All-Atom**

| Downsampling Factor | Channels/Layer | # Params [1e6] | C$\alpha$-RMSD (Å) ↓ | All-Atom RMSD (Å) ↓ | GDT-TS ↑ | GDT-HA ↑ | Residue Acc. (%) ↑ |
|---|---|---|---|---|---|---|---|
| 17 | [16, 24, 36] | 0.34 | 0.90 ± 0.20 | 0.68 ± 0.08 | 94 ± 3 | 76 ± 4 | 97 ± 2 |
| 17 | [16, 27, 45] | 0.38 | 0.89 ± 0.21 | 0.70 ± 0.09 | 94 ± 3 | 77 ± 5 | 98 ± 1 |
| 17 | [16, 32, 64] | 0.49 | 1.02 ± 0.25 | 0.72 ± 0.09 | 92 ± 4 | 73 ± 5 | 98 ± 1 |
| 53 | [16, 24, 36, 54] | 0.49 | 1.03 ± 0.18 | 0.83 ± 0.10 | 91 ± 3 | 72 ± 5 | 60 ± 4 |
| 53 | [16, 27, 45, 76] | 0.67 | 0.92 ± 0.19 | 0.77 ± 0.09 | 93 ± 3 | 75 ± 5 | 66 ± 4 |
| 53 | [16, 32, 64, 128] | 1.26 | 1.25 ± 0.32 | 0.80 ± 0.16 | 86 ± 5 | 65 ± 6 | 67 ± 5 |
| 160 | [16, 24, 36, 54, 81] | 0.77 | 1.67 ± 0.24 | 1.81 ± 0.16 | 77 ± 4 | 54 ± 5 | 17 ± 3 |
| 160 | [16, 27, 45, 76, 129] | 1.34 | 1.39 ± 0.23 | 1.51 ± 0.17 | 83 ± 4 | 60 ± 5 | 17 ± 3 |
| 160 | [16, 32, 64, 128, 256] | 3.77 | 1.21 ± 0.25 | 1.03 ± 0.15 | 87 ± 5 | 65 ± 6 | 27 ± 4 |

In our experiments we find a trade-off between domain coarsening factor and reconstruction performance. In Table 2 we show that although residue recovery suffers from large downsampling factors, structure recovery rates remain comparable between various settings. Moreover, we find that we are able to model all atoms in proteins, as opposed to only C$\alpha$ atoms (as commonly done), and still recover the structure with high precision. These results demonstrate that protein data can be captured effectively and efficiently using sequence-modular geometric representations. We directly utilize the learned compact latent space as shown below by various examples. For further ablation analysis please refer to Appendix D.

## 4.3 LATENT CONFORMATIONAL INTERPOLATION

To demonstrate the power of Ophiuchus's geometric latent space, we show smooth interpolation between two states of a protein structure without explicit latent regularization (as opposed to [Ramaswamy et al. (2021)]). We use the PDBFlex dataset [Hrabe et al. (2016)] and pick pairs of flexible proteins. Conformational snapshots of these proteins are used as the endpoints for the interpolation. We train a large Ophiuchus reconstruction model on general PDB data. The model coarsens up to

Table 3: Comparison to different diffusion models.

| Model | Dataset | Sampling Time (s) ↓ | scRMSD (< 2Å) ↑ | scTM (>0.5) ↑ | Diversity ↑ |
|---|---|---|---|---|---|
| FrameDiff [Yim et al. (2023)] | PDB | 8.6 | 0.17 | 0.81 | 0.42 |
| RFDiffusion [Trippe et al. (2023)] | PDB + AlphaFold DB | 50 | 0.79 | 0.99 | 0.64 |
| Ophiuchus-64 All-Atom | MiniProtein | 0.15 | 0.32 | 0.56 | 0.72 |
| Ophiuchus-485 Backbone | PDB | 0.46 | 0.18 | 0.36 | 0.39 |

485-residue proteins into a single high-order geometric representation using 6 convolutional down-sampling layers each with kernel size 5 and stride 3. The endpoint structures are compressed into single geometric representation which enables direct latent interpolation in feature .

We compare the results of linear interpolation in the latent space against linear interpolation in the coordinate domain (Fig. 11). To determine chemical validity of intermediate states, we scan protein data to quantify average bond lengths and inter-bond angles. We calculate the $L_2$ deviation from these averages for bonds and angles of interpolated structures. Additionally, we measure atomic clashes by counting collisions of Van der-Waals radii of non-bonded atoms (Fig. 11). Although the latent and autoencoder-reconstructed interpolations perform worse than direct interpolation near the original data points, we find that *only* the latent interpolation structures maintain a consistent profile of chemical validity throughout the trajectory, while direct interpolation in the coordinate domain disrupts it significantly. This demonstrates that the learned latent space compactly and smoothly represents protein conformations.

### 4.4 Latent Diffusion Experiments and Benchmarks

Our ablation study (Tables 2 and 4) shows successful recovery of backbone structure of large proteins even for large coarsening rates. However, we find that for sequence reconstruction, larger models and longer training times are required. During inference, all-atom models rely on the decoding of the sequence, thus making significantly harder for models to resolve all-atom generation. Due to computational constraints, we investigate all-atom latent diffusion models for short sequence lengths, and focus on backbone models for large proteins. We train the all-atom models with mini-proteins of sequences shorter than 64 residues, leveraging the MiniProtein scaffolds dataset produced by [Cao et al. (2022)]. In this regime, our model is precise and successfully reconstructs sequence and all-atom positions. We also instantiate an Ophiuchus model for generating the backbone trace for large proteins of length 485. For that, we train our model on the PDB data curated by [Yim et al. (2023)]. We compare the quality of diffusion samples from our model to RFDiffusion (T=50) and FrameDiff (N=500 and noise=0.1) samples of similar lengths. We generated 500 unconditional samples from each of the models for evaluation.

In Table 3 we compare sampling the latent space of Ophiuchus to existing models. We find that our model performs comparably in terms of different generated sample metrics, while enabling orders of magnitude faster sampling for proteins. For all comparisons we run all models on a single RTX6000 GPU. Please refer to Appendix F for more details.

### 5 Conclusion and Future Work

In this work, we introduced a new autoencoder model for protein structure and sequence representation. Through extensive ablation on its architecture, we quantified the trade-offs between model complexity and representation quality. We demonstrated the power of our learned representations in latent interpolation, and investigated its usage as basis for efficient latent generation of backbone and all-atom protein structures. Our studies suggest Ophiuchus to provide a strong foundation for constructing state-of-the-art protein neural architectures. In future work, we will investigate scaling Ophiuchus representations and generation to larger proteins and additional molecular domains.

### Reproducibility Statement

To reproduce the architectural components please refer to section 3 and appendix A. We will release architecture, training and experiments code upon acceptance.

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

APPENDIX

# A    ARCHITECTURE DETAILS

## A.1    ALL-ATOM REPRESENTATION

A canonical ordering of the atoms of each residue enables the local geometry to be described in a stacked array representation, where each feature channel corresponds to an atom. To directly encode positions, we stack the 3D coordinates of each atom. The coordinates vector behaves as the irreducible-representation of $SO(3)$ of degree $l = 1$. The atomic coordinates are taken relative to the $\mathbf{C}_\alpha$ of each residue. In practice, for implementing this ordering we follow the *atom14* tensor format of SidechainNet [King & Koes (2021)], where a vector $P \in \mathbb{R}^{14 \times 3}$ contains the atomic positions per residue. In Ophiuchus, we rearrange this encoding: one of those dimensions, the $\mathbf{C}_\alpha$ coordinate, is used as the absolute position; the 13 remaining 3D-vectors are centered at the $\mathbf{C}_\alpha$, and used as geometric features. The geometric features of residues with fewer than 14 atoms are zero-padded (Figure 4).

Still, four of the standard residues (Arginine, Glutamic Acid, Phenylalanine and Tyrosine) have at most two pairs of atoms that are interchangeable, due to the presence $180°$-rotation symmetries [Jumper et al. (2021)]. In Figure 5, we show how stacking their relative positions leads to representations that differ even when the same structure occurs across different rotations of a side-chain. To solve this issue, instead of stacking two 3D vectors $(\mathbf{P}_u^*, \mathbf{P}_v^*)$, our method uses a single $l = 1$ vector $\mathbf{V}_{\text{center}}^{l=1} = \frac{1}{2}(\mathbf{P}_v^* + \mathbf{P}_u^*)$. The mean makes this feature invariant to the $(v, u)$ atomic permutations, and the resulting vector points to the midpoint between the two atoms. To fully describe the positioning, difference $(\mathbf{P}_v^* - \mathbf{P}_u^*)$ must be encoded as well. For that, we use a single $l = 2$ feature $\mathbf{V}_{\text{diff}}^{l=2} = Y_{l=2}\left(\frac{1}{2}(\mathbf{P}_v^* - \mathbf{P}_u^*)\right)$. This feature is produced by projecting the difference of positions into a degree $l = 2$ spherical harmonics basis. Let $x \in \mathbb{R}^3$ denote a 3D vector. Then its projection into a feature of degree $l = 2$ is defined as:

$$Y_2(x) = \left[\sqrt{15} \cdot x_0 \cdot x_2, \sqrt{15} \cdot x0 \cdot x1, \frac{\sqrt{5}}{2}\left(-x_0^2 + 2x_1^2 - x_2^2\right), \sqrt{15} \cdot x_1 \cdot x_2, \frac{\sqrt{15}}{2} \cdot \left(-x_0^2 + x_2^2\right)\right]$$

Where each dimension of the resulting term is indexed by the order $m \in [-l, l] = [-2, 2]$, for degree $l = 2$. We note that for $m \in \{-2, -1, 1\}$, two components of $x$ directly multiply, while for $m \in \{0, 2\}$ only squared terms of $x$ are present. In both cases, the terms are invariant to flipping the sign of $x$, such that $Y_2(x) = Y_2(-x)$. Equivalently, $\mathbf{V}_{\text{diff}}^{l=2}$ is invariant to reordering of the two atoms $(v, u) \rightarrow (u, v)$:

$$\mathbf{V}_{\text{diff}}^{l=2} = Y_{l=2}\left(\frac{1}{2}(\mathbf{P}_v^* - \mathbf{P}_u^*)\right) = Y_{l=2}\left(\frac{1}{2}(\mathbf{P}_u^* - \mathbf{P}_v^*)\right)$$

In Figure 5, we compare the geometric latent space of a network that uses this permutation invariant encoding, versus one that uses naive stacking of atomic positions $\mathbf{P}^*$. We find that Ophiuchus correctly maps the geometry of the data, while direct stacking leads to representations that do not reflect the underlying symmetries.

## A.2    ALL-ATOM DECODING

Given a latent representation at the residue-level, $(\mathbf{P}, \mathbf{V}^{0:l_{\max}})$, we take $\mathbf{P}$ directly as the $\mathbf{C}_\alpha$ position of the residue. The hidden scalar representations $\mathbf{V}^{l=0}$ are transformed into categorical logits $\ell \in \mathbb{R}^{|\mathcal{R}|}$ to predict the probabilities of residue label $\hat{\mathbf{R}}$. To decode the side-chain atoms, Ophiuchus produces relative positions to $\mathbf{C}_\alpha$ for all atoms of each residue. During training, we enforce the residue label to be the ground truth. During inference, we output the residue label corresponding to the largest logit value.

Relative coordinates are produced directly from geometric features. We linearly project $\mathbf{V}^{l=1}$ to obtain the relative position vectors of orderable atoms $\hat{\mathbf{V}}_{\text{ordered}}^{l=1}$. To decode positions $(\hat{\mathbf{P}}_v^*, \hat{\mathbf{P}}_u^*)$ for an unorderable pair of atoms $(v, u)$, we linearly project $\mathbf{V}^{l>0}$ to predict $\hat{\mathbf{V}}_{\text{center}}^{l=1}$ and $\hat{\mathbf{V}}_{\text{diff}}^{l=2}$. To produce two relative positions out of $\hat{\mathbf{V}}_{\text{diff}}^{l=2}$, we determine the rotation axis around which the feature rotates the least by taking the left-eigenvector with smallest eigenvalue of $\mathcal{X}^{l=2}\hat{\mathbf{V}}_{\text{diff}}^{l=2}$, where $\mathcal{X}^{l=2}$ is the

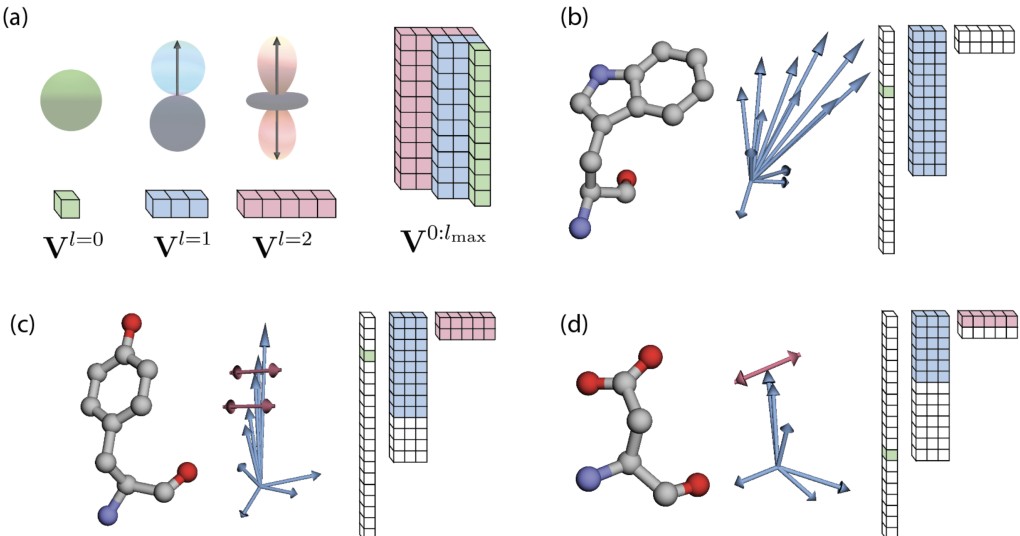

Figure 4: **Geometric Representations of Protein Residues.** **(a)** We leverage the bases of spherical harmonics to represent signals on SO(3). **(b-d)** Examples of individual protein residues encoded in spherical harmonics bases: residue label is directly encoded in a order $l = 0$ representation as a one-hot vector; atom positions in a canonical ordering are encoded as $l = 1$ features; additionally, unorderable atom positions are encoded as $l = 1$ and $l = 2$ features that are invariant to their permutation flips. In the figures, we displace $l = 2$ features for illustrative purposes – in practice, the signal is processed as centered in the $\mathbf{C}_\alpha$. **(b)** Tryptophan is the largest residue we consider, utilizing all dimensions of $n_{\max} = 13$ atoms in the input $\mathbf{V}^{l=1}_{\text{ordered}}$. **(c)** Tyrosine needs padding for $\mathbf{V}^{l=1}$, but produces two $\mathbf{V}^{l=2}$ from its two pairs of permutable atoms. **(d)** Aspartic Acid has a single pair of permutable atoms, and its $\mathbf{V}^{l=2}$ is padded.

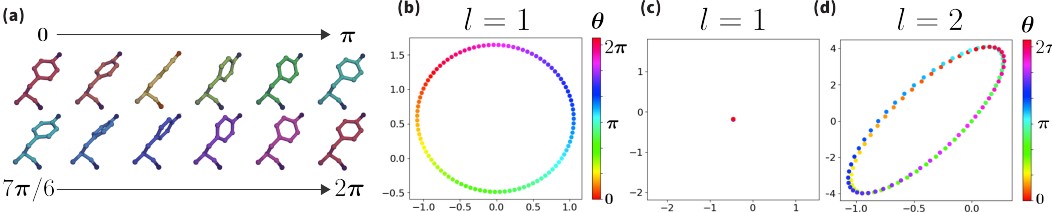

Figure 5: **Permutation Invariance of Side-chain Atoms in Stacked Geometric Representations.** **(a)** We provide an example in which we rotate the side-chain of Tyrosine by $2\pi$ radians while keeping the ordering of atoms fixed. Note that the structure is the same after rotation by $\pi$ radians. **(b)** An SO(3)-equivariant network may stack atomic relative positions in a canonical order. However, because of the permutation symmetries, the naive stacked representation will lead to latent representations that are different even when the data is geometrically the same. To demonstrate this, we plot two components ($m = -1$ and $m = 0$) of an internal $\mathbf{V}^{l=1}$ feature, while rotating the side-chain positions by $2\pi$ radians. This network represents the structures rotated by $\theta = 0$ (red) and $\theta = \pi$ (cyan) differently despite having exactly the same geometric features. **(c-d)** Plots of internal $\mathbf{V}^{l=1,2}$ of Ophiuchus which encodes the position of permutable pairs jointly as an $l = 1$ center of symmetry and an $l = 2$ difference of positions, resulting in the *same* representation for structures rotated by $\theta = 0$ (red) and $\theta = \pi$ (cyan).

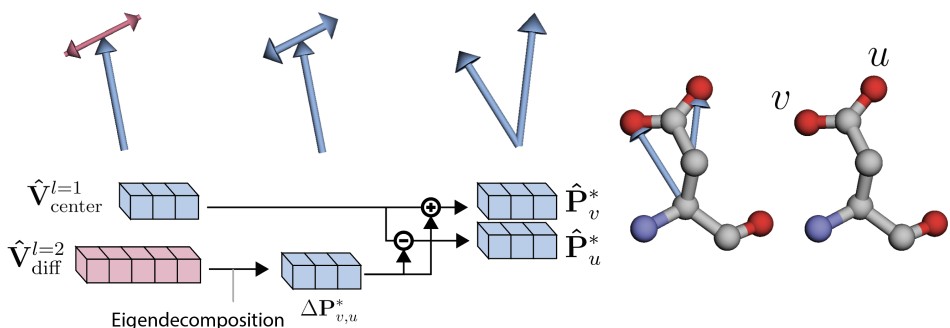

Figure 6: **Decoding Permutable Atoms.** Sketch on how to decode a double-sided arrow ($\mathbf{V}^{l=2}$ signals) into two unordered vectors ($\hat{\mathbf{P}}_u^*, \hat{\mathbf{P}}_v^*$). $Y^{l=2} : \mathbb{R}^3 \to \mathbb{R}^5$ can be viewed as a 3 dimensional manifold embed in a 5 dimensional space. We exploit the fact that the points on that manifold are unaffected by a rotation around their pre-image vector. To extend the definition to all the points of $\mathbb{R}^5$ (i.e. also outside of the manifold), we look for the axis of rotation with the smallest impact on the 5d entry. For that, we compute the left-eigenvector with the smallest eigenvalue of the action of the generators on the input point: $\mathcal{X}^{l=2}\hat{\mathbf{V}}_{\mathrm{diff}}^{l=2}$. The resulting axis is used as a relative position $\pm\Delta\mathbf{P}_{v,u}^*$ between the two atoms, and is used to recover atomic positions through $\mathbf{P}_v^* = \mathbf{V}_{\mathrm{center}}^{l=1} + \Delta\mathbf{P}_{v,u}^*$ and $\mathbf{P}_u^* = \mathbf{V}_{\mathrm{center}}^{l=1} - \Delta\mathbf{P}_{v,u}^*$.

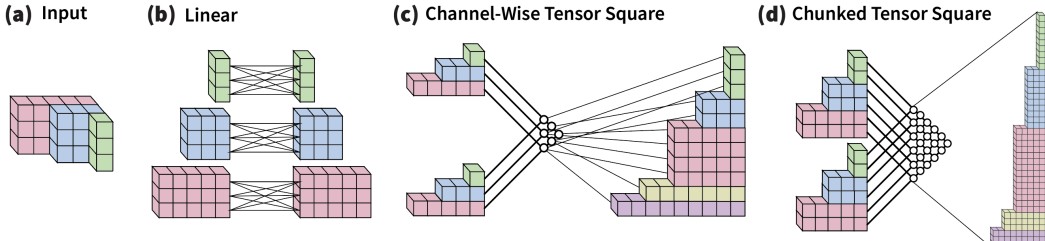

Figure 7: **Building Blocks of Self-Interaction. (a)** Self-Interaction updates only SO(3)-equivariant features, which are represented as a $D$ channels each with vectors of degree $l$ up to degree $l_{\mathrm{max}}$. **(b)** A roto-translational equivariant linear layer transforms only within the same order $l$. **(c-d)** we use the tensor square operation to interact features across different degrees $l$. We employ two instantiations of this operation. **(c)** The Self-Interaction in autoencoder models applies the square operation within the same channel of the representation, interacting features with themselves across $l$ of same channel dimension $d \in [0, D]$. **(d)** The Self-Interaction in diffusion models chunks the representation in groups of channels before the square operation. It is more expressive, but imposes a harder computational cost.

generator of the irreducible representations of degree $l = 2$. We illustrate this procedure in Figure 6 and explain the process in detail in the caption. This method proves effective because the output direction of this axis is inherently ambiguous, aligning perfectly with our requirement for the vectors to be unorderable.

## A.3 DETAILS OF SELF-INTERACTION

The objective of our Self-Interaction module is to function exclusively based on the geometric features $\mathbf{V}^{0:l_{\mathrm{max}}}$, while concurrently mixing irreducible representations across various $l$ values. To accomplish this, we calculate the tensor product of the representation with itself; this operation is termed the "tensor square" and is symbolized by $\left(\mathbf{V}^{0:l_{\mathrm{max}}}\right)^{\otimes 2}$. As the channel dimensions expand, the computational complexity tied to the tensor square increases quadratically. To solve this computational load, we instead perform the square operation channel-wise, or by chunks of channels. Figures 7.c and 7.d illustrate these operations. After obtaining the squares from the chunked or

**(a) Spatial Convolution**

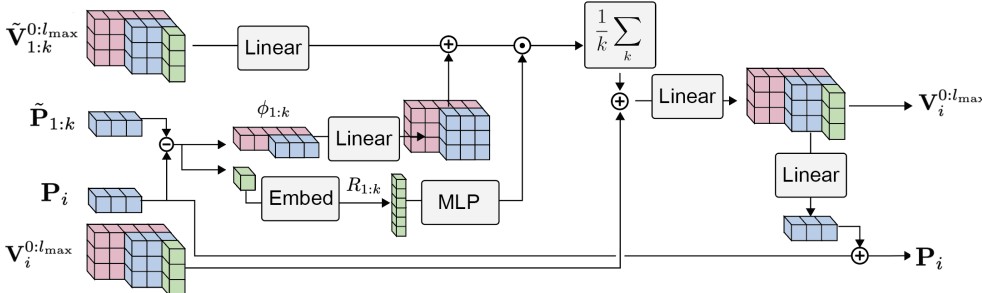

**(b) Sequence Convolution**

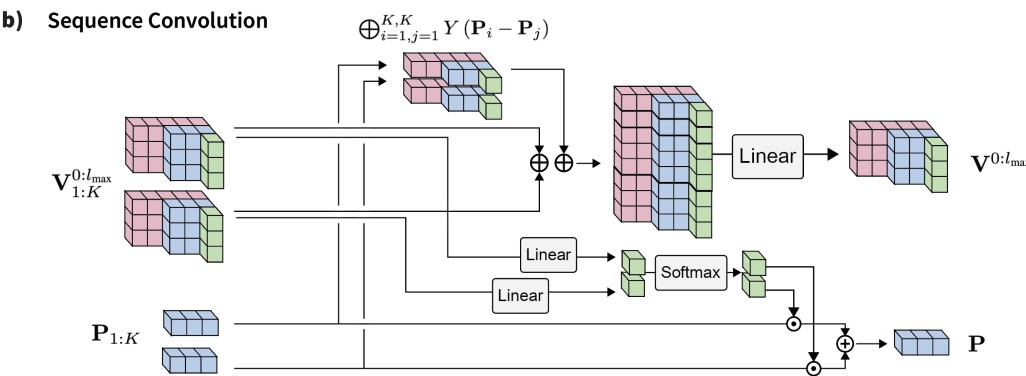

Figure 8: **Convolutions of Ophiuchus.** **(a)** In the Spatial Convolution, we update the feature representation $\mathbf{V}_i^{0:l_{\max}}$ and position $\mathbf{P}_i$ of a node $i$ by first aggregating messages from its $k$ nearest-neighbors. The message is composed out of the neighbor features $\tilde{\mathbf{V}}_{1:k}^{0:l_{\max}}$ and the relative position between the nodes $\mathbf{P}_i - \tilde{\mathbf{P}}_{1:k}$. After updating the features $\mathbf{V}_i^{0:l_{\max}}$, we project vectors $\mathbf{V}_i^{l=1}$ to predict an update to the position $\mathbf{P}_i$. **(b)** In the Sequence Convolution, we concatenate the feature representations of sequence neighbors $\tilde{\mathbf{V}}_{1:K}^{0:l_{\max}}$ along with spherical harmonic signals of their relative positions $\mathbf{P}_i - \mathbf{P}_j$, for $i \in [1, K]$, $j \in [1, K]$. The concatenated feature vector is then linearly projected to the output dimensionality. To produce a coarsened position, each $\tilde{\mathbf{V}}_{1:K}^{0:l_{\max}}$ produces a score that is used as weight in summing the original positions $\mathbf{P}_{1:K}$. The Softmax function is necessary to ensure the sum of the weights is normalized.

individual channels, the segmented results are subsequently concatenated to generate an updated representation $\mathbf{V}^{0:l_{\max}}$, which is transformed through a learnable linear layer to the output dimensionality.

## A.4 NON-LINEARITIES

To incorporate non-linearities in our geometric representations $\mathbf{V}^{l=0:l_{\max}}$, we employ a similar roto-translation equivariant gate mechanism as described in Equiformer [Liao & Smidt (2023)]. This mechanism is present at the last step of Self-Interaction and in the message preparation step of Spatial Convolution (Figure (2)). In both cases, we implement the activation by first isolating the scalar representations $\mathbf{V}^{l=0}$ and transforming them through a standard MultiLayerPerceptron (MLP). We use the SiLu activation function [Elfwing et al. (2017)] after each layer of the MLP. In the output vector, a scalar is produced for and multiplied into each channel of $\mathbf{V}^{l=0:l_{\max}}$.

## A.5 Roto-Translation Equivariance of Sequence Convolution

A Sequence Convolution kernel takes in $K$ coordinates $\mathbf{P}_{1:K}$ to produce a single coordinate $\mathbf{P} = \sum_{i=1}^{K} w_i \mathbf{P}_i$. We show that these weights need to be normalized in order for translation equivariance to be satisfied. Let $T$ denote a 3D translation vector, then translation equivariance requires:

$$(\mathbf{P} + T) = \sum_{i=1}^{K} w_i (\mathbf{P}_i + T) = \sum_{i=1}^{K} w_i \mathbf{P}_i + \sum_{i=1}^{K} w_i T \rightarrow \sum_{i=1}^{K} w_i = 1$$

Rotation equivariance is immediately satisfied since the sum of 3D vectors is a rotation equivariant operation. Let $R$ denote a rotation matrix. Then,

$$(R\mathbf{P}) = R \sum_{i=1}^{K} w_i (\mathbf{P}_i) = \sum_{i=1}^{K} w_i (R\mathbf{P}_i)$$

## A.6 Transpose Sequence Convolution

Given a single coarse anchor position and features representation $(\mathbf{P}, \mathbf{V}^{l=0:l_{\max}})$, we first map $\mathbf{V}^{l=0:l_{\max}}$ into a new representation, reshaping it by chunking $K$ features. We then project $\mathbf{V}^{l=1}$ and produce $K$ relative position vectors $\Delta \mathbf{P}_{1:K}$, which are summed with the original position $\mathbf{P}$ to produce $K$ new coordinates.

---

**Algorithm 6:** Transpose Sequence Convolution

---

**Input:** Kernel Size $K$
**Input:** Latent Representations $(\mathbf{P}, \mathbf{V}^{0:l_{\max}})$
$\Delta \mathbf{P}_{1:K} \leftarrow \text{Linear}\left(\mathbf{V}^{l=1}\right)$           ▷ `Predict` $K$ `relative positions`
$\mathbf{P}_{1:K} \leftarrow \mathbf{P} + \Delta \mathbf{P}_{1:K}$
$\mathbf{V}^{l=0:l_{\max}}_{1:K} \leftarrow \text{Reshape}_K\left(\text{Linear}(\mathbf{V}^{l=0:l_{\max}})\right)$    ▷ `Split the channels in` $K$ `chunks`
**return** $(\mathbf{P}, \mathbf{V}^{0:l_{\max}})_{1:K}$

---

This procedure generates windows of $K$ representations and positions. These windows may intersect in the decoded output. We resolve those superpositions by taking the average position and average representations within intersections.

## A.7 Layer Normalization and Residual Connections

Training deep models can be challenging due to vanishing or exploding gradients. We employ layer normalization [Ba et al. (2016)] and residual connections [He et al. (2015)] in order to tackle those challenges. We incorporate layer normalization and residual connections at the end of every Self-Interaction and every convolution. To keep roto-translation equivariance, we use the layer normalization described in Equiformer [Liao & Smidt (2023)], which rescales the $l$ signals independetly within a representation by using the root mean square value of the vectors. We found both residuals and layer norms to be critical in training deep models for large proteins.

## A.8 ENCODER-DECODER

Below we describe the encoder/decoder algorithm using the building blocks previously introduced.

---
**Algorithm 7:** Encoder

---
**Input:** $C_\alpha$ Position $\mathbf{P}^\alpha \in \mathbb{R}^{1 \times 3}$
**Input:** All-Atom Relative Positions $\mathbf{P}^* \in \mathbb{R}^{n \times 3}$
**Input:** Residue Label $\mathbf{R} \in \mathcal{R}$
**Output:** Latent Representation $(\mathbf{P}, \mathbf{V}^{l=0:l_{\max}})$
$(\mathbf{P}, \mathbf{V}^{0:2}) \leftarrow$ All-Atom Encoding$(\mathbf{P}^\alpha, \mathbf{P}^*, \mathbf{R})$       ▷ Initial Residue Encoding
$(\mathbf{P}, \mathbf{V}^{0:l_{\max}}) \leftarrow$ Linear$(\mathbf{P}, \mathbf{V}^{0:2})$
**for** $i \leftarrow 1$ **to** $L$ **do**
     $(\mathbf{P}, \mathbf{V}^{0:l_{\max}}) \leftarrow$ Self-Interaction$(\mathbf{P}, \mathbf{V}^{0:l_{\max}})$
     $(\mathbf{P}, \mathbf{V}^{0:l_{\max}}) \leftarrow$ Spatial Convolution$(\mathbf{P}, \mathbf{V}^{0:l_{\max}})$
     $(\mathbf{P}, \mathbf{V}^{0:l_{\max}}) \leftarrow$ Sequence Convolution$(\mathbf{P}, \mathbf{V}^{0:l_{\max}})$
**end**
**return** $(\mathbf{P}, \mathbf{V}^{0:l_{\max}})$

---

---
**Algorithm 8:** Decoder

---
**Input:** Latent Representation $(\mathbf{P}, \mathbf{V}^{0:l_{\max}})$
**Output:** Protein Structure and Sequence Logits $(\hat{\mathbf{P}}^\alpha, \hat{\mathbf{P}}^*, \boldsymbol{\ell})$
**for** $i \leftarrow 1$ **to** $L$ **do**
     $(\mathbf{P}, \mathbf{V}^{0:l_{\max}}) \leftarrow$ Self-Interaction$(\mathbf{P}, \mathbf{V}^{0:l_{\max}})$
     $(\mathbf{P}, \mathbf{V}^{0:l_{\max}}) \leftarrow$ Spatial Convolution$(\mathbf{P}, \mathbf{V}^{0:l_{\max}})$
     $(\mathbf{P}, \mathbf{V}^{0:l_{\max}}) \leftarrow$ Transpose Sequence Convolution$(\mathbf{P}, \mathbf{V}^{0:l_{\max}})$
**end**
$(\mathbf{P}, \mathbf{V}^{0:2}) \leftarrow$ Linear$(\mathbf{P}, \mathbf{V}^{0:l_{\max}})$
$(\hat{\mathbf{P}}^\alpha, \hat{\mathbf{P}}^*, \boldsymbol{\ell}) \leftarrow$ All-Atom Decoding$(\mathbf{P}, \mathbf{V}^{0:2})$       ▷ Final Protein Decoding
**return** $(\hat{\mathbf{P}}^\alpha, \hat{\mathbf{P}}^*, \boldsymbol{\ell})$

---

## A.9 VARIABLE SEQUENCE LENGTH

We handle proteins of different sequence length by setting a maximum size for the model input. During training, proteins that are larger than the maximum size are cropped, and those that are smaller are padded. The boundary residue is given a special token that labels it as the end of the protein. For inference, we crop the tail of the output after the sequence position where this special token is predicted.

## A.10 TIME COMPLEXITY

We analyse the time complexity of a forward pass through the Ophiuchus autoencoder. Let the list $(\mathbf{P}, \mathbf{V}^{0:l_{\max}})_N$ be an arbitrary latent state with $N$ positions and $N$ geometric representations of dimensionality $D$, where for each $d \in [0, D]$ there are geometric features of degree $l \in [0, l_{\max}]$. Note that size of a geometric representation grows as $O(D \cdot l_{\max}^2)$ in memory.

- The cost of an SO(3)-equivariant linear layer (Fig. 7.b) is $O(D^2 \cdot l_{\max}^3)$.

- The cost of a channel-wise tensor square operation (Fig. 7.c) is $O(D \cdot l_{\max}^4)$.

- In **Self-Interaction** (Alg. 3), we use a tensor square and project it using a linear layer. The time complexity is given by $O\left(N \cdot (D^2 \cdot l_{\max}^3 + D \cdot l_{\max}^4)\right)$ for the whole protein.

- In **Spatial Convolution** (Alg. 5), a node aggregates geometric messages from $k$ of its neighbors. Resolving the $k$ nearest neighbors can be efficiently done in $O((N + k) \log N)$ through the k-d data structure [Bentley (1975)]. For each residue, its $k$ neighbors prepare messages through linear layers, at total cost $O(N \cdot k \cdot D^2 \cdot l_{\max}^3)$.

- In a **Sequence Convolution** (Alg. 4), a kernel stacks $K$ geometric representations of dimensionality $D$ and linearly maps them to a new feature of dimensionality $\rho \cdot D$, where $\rho$ is a rescaling factor, yielding $O((K \cdot D) \cdot (\rho \cdot D) \cdot l_{\max}^3) = O(K \cdot \rho \cdot D^2 \cdot l_{\max}^3)$. With length $N$ and stride $S$, the total cost is $O\left(\frac{N}{S} \cdot K \cdot \rho \cdot D^2 \cdot l_{\max}^3\right)$.

- The cost of an **Ophiuchus Block** is the sum of the terms above,

$$O\left(D \cdot l_{\max}^3 \cdot N \cdot (\frac{K}{S} \cdot \rho D + kD)\right)$$

- An **Autoencoder** (Alg. 7,8) that uses stride $S$ convolutions for coarsening uses $L = O\left(\log_S(N)\right)$ layers to reduce a protein of size $N$ into a single representation. At depth $i$, the dimensionality is given by $D_i = \rho^i D$ and the sequence length is given by $N_i = \frac{N}{S^i}$. The time complexity of our Autoencoder is given by geometric sum:

$$O\left(\sum_i^{\log_S(N)}\left(l_{\max}^3 \rho^i D \frac{N}{S^i}(K\rho^{i+1}D + k\rho^i D)\right)\right) = O\left(Nl_{\max}^3(K\rho + k)D^2 \sum_i^{\log_S(N)}\left(\frac{\rho^2}{S}\right)^i\right)$$

We are interested in the dependence on the length $N$ of a protein, therefore, we keep only relevant parameters. Summing the geometric series and using the identity $x^{\log_b(a)} = a^{\log_b(x)}$ we get:

$$= O\left(N\frac{(\frac{\rho^2}{S})^{\log_S(N)+1} - 1}{\frac{\rho^2}{S} - 1}\right) = \begin{cases} O\left(N^{1+\log_S \rho^2}\right) & \text{for } \rho^2/S > 1 \\ O\left(N\log_S N\right) & \text{for } \rho^2/S = 1 \\ O\left(N\right) & \text{for } \rho^2/S < 1 \end{cases}$$

In most of our experiments we operate in the $\rho^2/S < 1$ regime.

## B  LOSS DETAILS

### B.1  DETAILS ON VECTOR MAP LOSS

The Huber Loss [Huber (1992)] behaves linearly for large inputs, and quadratically for small ones. It is defined as:

$$\text{HuberLoss}(y, f(x)) = \begin{cases} \frac{1}{2}(y - f(x))^2 & \text{if } |y - f(x)| \leq \delta, \\ \delta \cdot |y - f(x)| - \frac{1}{2}\delta^2 & \text{otherwise.} \end{cases}$$

We found it to significantly improve training stability for large models compared to mean squared error. We use $\delta = 0.5$ for all our experiments.

The vector map loss measures differences of internal vector maps $V(\mathbf{P}) - V(\hat{\mathbf{P}})$ between predicted and ground positions, where $V(\mathbf{P})^{i,j} = (\mathbf{P}^i - \mathbf{P}^j) \in \mathbb{R}^{\times 3}$. Our output algorithm for decoding atoms produces arbitrary symmetry breaks (Appendix A.2) for positions $\mathbf{P}_v^*$ and $\mathbf{P}_u^*$ of atoms that are not orderable. Because of that, a loss on the vector map is not directly applicable to the output of our model, since the order of the model output might differ from the order of the ground truth data. To solve that, we consider both possible orderings of permutable atoms, and choose the one that minimizes the loss. Solving for the optimal ordering is not feasible for the system as a whole, since the number of permutations to be considered scales exponentially with $N$. Instead, we first compute a vector map loss internal to each residue. We consider the alternative order of permutable atoms, and choose the candidate that minimizes this local loss. This ordering is used for the rest of our losses.

### B.2  CHEMICAL LOSSES

We consider bonds, interbond angles, dihedral angles and steric clashes when computing a loss for chemical validity. Let $\mathbf{P} \in \mathbb{R}^{N_{\text{Atom}} \times 3}$ be the list of atom positions in ground truth data. We denote $\hat{\mathbf{P}} \in \mathbb{R}^{N_{\text{Atom}} \times 3}$ as the list of atom positions predicted by the model. For each chemical interaction, we

precompute indices of atoms that perform the interaction. For example, for bonds we precompute a list of pairs of atoms that are bonded according to the chemical profile of each residue. Our chemical losses then take form:

$$\mathcal{L}_{\text{Bonds}} = \frac{1}{|\mathcal{B}|} \sum_{(v,u)\in\mathcal{B}} \left\| \; ||\hat{\mathbf{P}}_v - \hat{\mathbf{P}}_u||_2 - ||\mathbf{P}_v - \mathbf{P}_u||_2 \; \right\|_2^2$$

where $\mathcal{B}$ is a list of pair of indices of atoms that form a bond. We compare the distance between bonded atoms in prediction and ground truth data.

$$\mathcal{L}_{\text{Angles}} = \frac{1}{|\mathcal{A}|} \sum_{(v,u,p)\in\mathcal{A}} ||\alpha(\hat{\mathbf{P}}_v, \hat{\mathbf{P}}_u \hat{\mathbf{P}}_p) - \alpha(\mathbf{P}_v, \mathbf{P}_u, \mathbf{P}_p)||_2^2$$

where $\mathcal{A}$ is a list of 3-tuples of indices of atoms that are connected through bonds. The tuple takes the form $(v, u, p)$ where $u$ is connected to $v$ and to $p$. Here, the function $\alpha(\cdot, \cdot, \cdot)$ measures the angle in radians between positions of atoms that are connected through bonds.

$$\mathcal{L}_{\text{Dihedrals}} = \frac{1}{|\mathcal{D}|} \sum_{(v,u,p,q)\in\mathcal{D}} ||\tau(\hat{\mathbf{P}}_v, \hat{\mathbf{P}}_u, \hat{\mathbf{P}}_p, \hat{\mathbf{P}}_q) - \tau(\mathbf{P}_v, \mathbf{P}_u, \mathbf{P}_p, \mathbf{P}_q)||_2^2$$

where $\mathcal{D}$ is a list of 4-tuples of indices of atoms that are connected by bonds, that is, $(v, u, p, q)$ where $(v, u)$, $(u, p)$ and $(p, q)$ are connected by bonds. Here, the function $\tau(\cdot, \cdot, \cdot, \cdot)$ measures the dihedral angle in radians.

$$\mathcal{L}_{\text{Clashes}} = \frac{1}{|\mathcal{C}|} \sum_{(v,u)\in\mathcal{C}} H(r_v + r_u - ||\hat{\mathbf{P}}_v - \hat{\mathbf{P}}_u||_2)$$

where $H$ is a smooth and differentiable Heaviside-like step function, $\mathcal{C}$ is a list of pair of indices of atoms that are not bonded, and $(r_v, r_u)$ are the van der Waals radii of atoms $v, u$.

## B.3 REGULARIZATION

When training autoencoder models for latent diffusion, we regularize the learned latent space so that representations are amenable to the relevant range scales of the source distribution $\mathcal{N}(0, 1)$. Let $\mathbf{V}_i^l$ denote the $i$-th channel of a vector representation $\mathbf{V}^l$. We regularize the autoencoder latent space by optimizing radial and angular components of our vectors:

$$\mathcal{L}_{\text{reg}} = \sum_i \text{ReLU}(1 - ||\mathbf{V}_i^l||_1^1) + \sum_i \sum_{i\neq j}(\mathbf{V}_i^l \cdot \mathbf{V}_j^l)$$

The first term penalizes vector magnitudes larger than one, and the second term induces vectors to spread angularly. We find these regularizations to significantly help training of the denoising diffusion model.

## B.4 TOTAL LOSS

We weight the different losses in our pipeline. For the standard training of the autoencoder, we use weights:

$$\mathcal{L} = 10 \cdot \mathcal{L}_{\text{VectorMap}} + \mathcal{L}_{\text{CrossEntropy}} + 0.1 \cdot \mathcal{L}_{\text{Bonds}} + 0.1 \cdot L_{\text{Angles}} + 0.1 \cdot \mathcal{L}_{\text{Dihedrals}} + 10 \cdot \mathcal{L}_{\text{Clashes}}$$

For fine-tuning the model, we increase the weight of chemical losses significantly:

$$\mathcal{L} = 10 \cdot \mathcal{L}_{\text{VectorMap}} + \mathcal{L}_{\text{CrossEntropy}} + 1.0 \cdot \mathcal{L}_{\text{Bonds}} + 1.0 \cdot L_{\text{Angles}} + 1.0 \cdot \mathcal{L}_{\text{Dihedrals}} + 100 \cdot \mathcal{L}_{\text{Clashes}}$$

We find that high weight values for chemical losses at early training may hurt the model convergence, in particular for models that operate on large lengths.

## C  DETAILS ON AUTOENCODER COMPARISON

We implement the comparison EGNN model in its original form, following [Satorras et al. (2022)]. We use kernel size $K = 3$ and stride $S = 2$ for downsampling and upsampling, and follow the procedure described in [Fu et al. (2023)]. For this comparison, we train the models to minimize the residue label cross entropy, and the vector map loss.

Ophiuchus significantly outperforms the EGNN-based architecture. The standard EGNN is *SO(3)-equivariant* with respect to its positions, however it models features with *SO(3)-invariant* representations. As part of its message passing, EGNN uses relative vectors between nodes to update positions. However, when downsampling positions, the total number of relative vectors available reduces quadratically, making it increasingly challenging to recover coordinates. Our method instead uses *SO(3)-equivariant* feature representations, and is able to keep 3D vector information in features as it coarsens positions. Thus, with very few parameters our model is able to encode and recover protein structures.

## D  MORE ON ABLATION

In addition to all-atom ablation found in Table 2 we conducted a similar ablation study on models trained on backbone only atoms as shown in Table 4. We found that backbone only models performed slightly better on backbone reconstruction. Furthermore, in Fig. 9.a we compare the relative vector loss between ground truth coordinates and the coordinates reconstructed from the autoencoder with respect to different downsampling factors. We average over different trials and channel rescale factors. As expected, we find that for lower downsampling factors the structure reconstruction accuracy is better. In Fig 9.b we similarly plot the residue recovery accuracy with respect to different downsampling factors. Again, we find the expected result, the residue recovery is better for lower downsampling factors.

Notably, the relative change in structure recovery accuracy with respect to different downsampling factors is much lower compared to the relative change in residue recovery accuracy for different downsampling factors. This suggest our model was able to learn much more efficiently a compressed prior for structure as compared to sequence, which coincides with the common knowledge that sequence has high redundancy in biological proteins. In Fig. 9.c we compare the structure reconstruction accuracy across different channel rescaling factors. Interestingly we find that for larger rescaling factors the structure reconstruction accuracy is slightly lower.

However, since we trained only for 10 epochs, it is likely that due to the larger number of model parameters when employing a larger rescaling factor it would take somewhat longer to achieve similar results. Finally, in Fig. 9.d we compare the residue recovery accuracy across different rescaling factors. We see that for higher rescaling factors we get a higher residue recovery rate. This suggests that sequence recovery is highly dependant on the number of model parameters and is not easily capturable by efficient structural models.

Table 4: **Recovery rates from bottleneck representations - Backbone only**

| Factor | Channels/Layer | # Params [1e6] ↓ | Cα-RMSD (Å) ↓ | GDT-TS ↑ | GDT-HA ↑ |
|---|---|---|---|---|---|
| 17 | [16, 24, 36] | 0.34 | 0.81 ± 0.31 | 96 ± 3 | 81 ± 5 |
| 17 | [16, 27, 45] | 0.38 | 0.99 ± 0.45 | 95 ± 3 | 81 ± 6 |
| 17 | [16, 32, 64] | 0.49 | 1.03 ± 0.42 | 92 ± 4 | 74 ± 6 |
| 53 | [16, 24, 36, 54] | 0.49 | 0.99 ± 0.38 | 92 ± 5 | 74 ± 8 |
| 53 | [16, 27, 45, 76] | 0.67 | 1.08 ± 0.40 | 91 ± 6 | 71 ± 8 |
| 53 | [16, 32, 64, 128] | 1.26 | 1.02 ± 0.64 | 92 ± 9 | 75 ± 11 |
| 160 | [16, 24, 36, 54, 81] | 0.77 | 1.33 ± 0.42 | 84 ± 7 | 63 ± 8 |
| 160 | [16, 27, 45, 76, 129] | 1.34 | 1.11 ± 0.29 | 89 ± 4 | 69 ± 7 |
| 160 | [16, 32, 64, 128, 256] | 3.77 | 0.90 ± 0.44 | 94 ± 7 | 77 ± 9 |

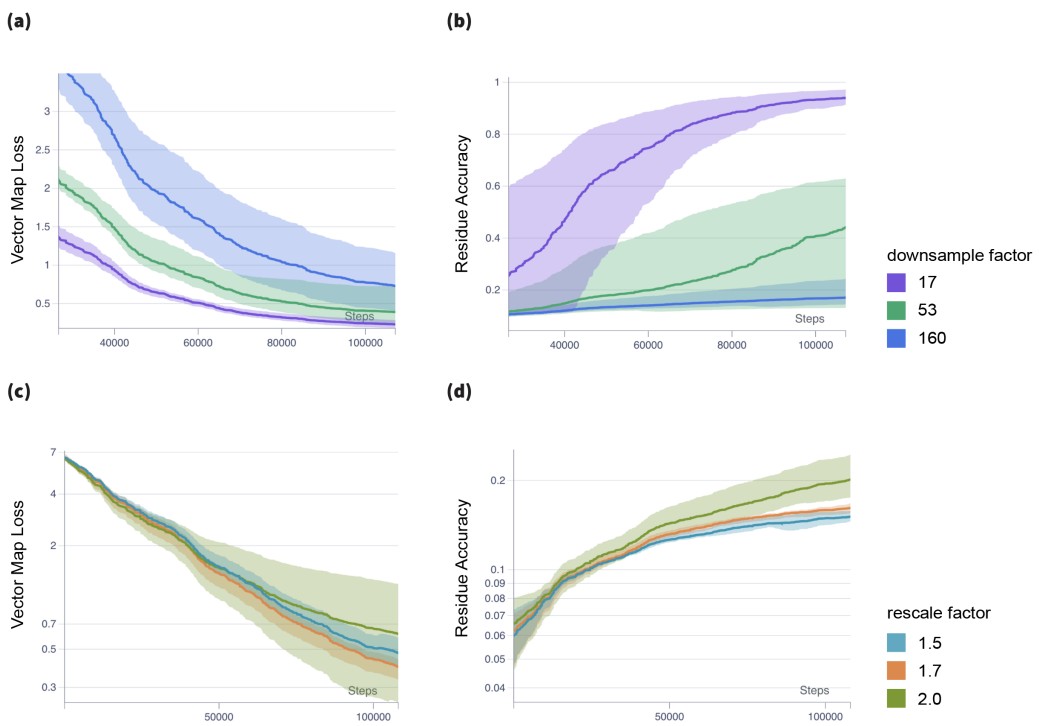

Figure 9: **Ablation Training Curves.** We plot metrics across 10 training epochs for our ablated models from Table 2. **(a-b)** compares models across downsampling factors and highlights the trade-off between downsampling and reconstruction. **(c-d)** compares different rescaling factors for fixed downsampling at 160.

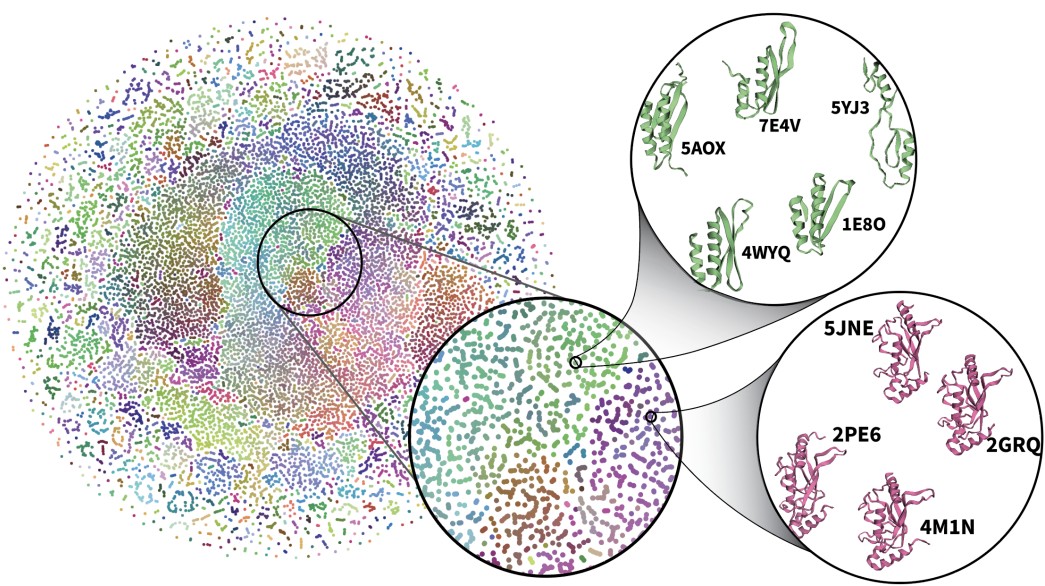

Figure 10: **Latent Space Analysis**: we qualitatively examine the unsupervised clustering capabilities of our representations through t-SNE.

# E    LATENT SPACE ANALYSIS

## E.1    VISUALIZATION OF LATENT SPACE

To visualize the learned latent space of Ophiuchus, we forward 50k samples from the training set through a large 485-length model (Figure 10). We collect the scalar component of bottleneck representations, and use t-SNE to produce 2D points for coordinates. We similarly produce 3D points and use those for coloring. The result is visualized in Figure 10. Visual inspection of neighboring points reveals unsupervised clustering of similar folds and sequences.

# F    LATENT DIFFUSION DETAILS

We train all Ophiuchus diffusion models with learning rate lr $= 1 \times 10^{-3}$ for 10,000 epochs. For denoising networks we use Self-Interactions with the chunked-channel tensor square operation (Alg. 3).

Our tested models are trained on two different datasets. The MiniProtein scaffolds dataset consists of 66k all-atom structures of sequence length between 50 and 65, and composes of diverse folds and sequences across 5 secondary structure topologies, and is introduced by [Cao et al. (2022)]. We also train a model on the data curated by [Yim et al. (2023)], which consists of approximately 22k proteins, to compare Ophiuchus to the performance of FrameDiff and RFDiffusion in backbone generation for proteins up to 485 residues.

## F.1    SELF-CONSISTENCY SCORES

To compute the scTM scores, we recover 8 sequences using ProteinMPNN for 500 sampled backbones from the tested diffusion models. We used a sampling temperature of 0.1 for ProteinMPNN. Unlike the original work, where the sequences where folded using AlphaFold2, we use OmegaFold [Wu et al. (2022b)] similar to [Lin & AlQuraishi (2023)]. The predicted structures are aligned to the original sampled backbones and TM-Score and RMSD is calculated for each alignment. To calculate the diversity measurement, we hierarchically clustered samples using MaxCluster. Diversity is defined as the number of clusters divided by the total number of samples, as described in [Yim et al. (2023)].

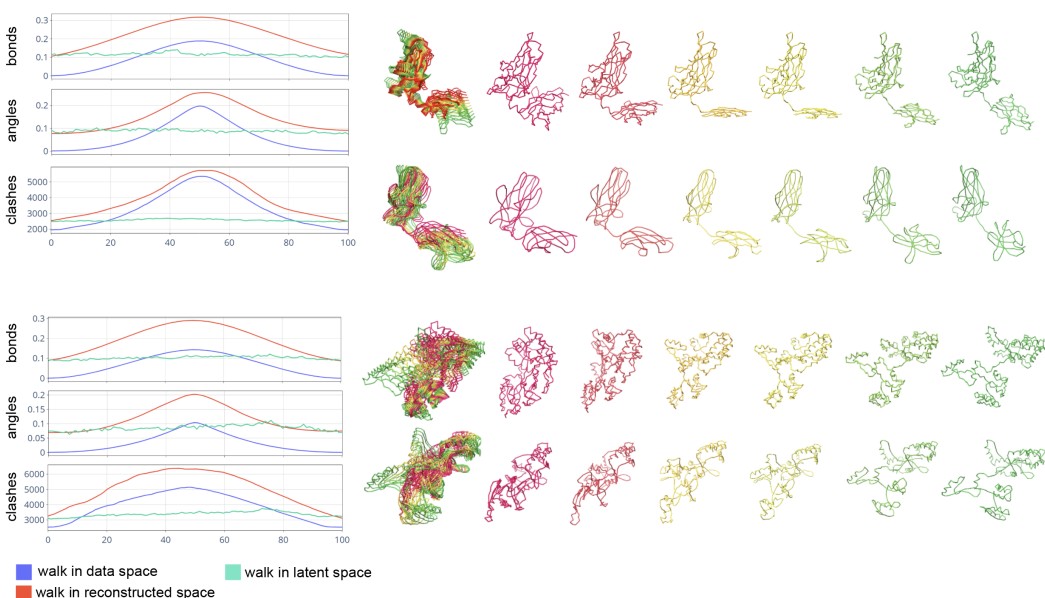

Figure 11: **Latent Conformational Interpolation. Top:** Comparison of chemical validity metrics across interpolated structures of Nuclear Factor of Activated T-Cell (NFAT) Protein (PDB ID: 1S9K and PDB ID: 2O93). **Bottom:** Comparison of chemical validity metrics across interpolated structures of Maltose-binding periplasmic protein (PDB ID: 4JKM and PDB 6LF3). For both proteins, we plot results for interpolations on the original data space, on the latent space, and on the autoencoder-reconstructed data space.

## F.2 MiniProtein Model

In Figure (13) we show generated samples from our miniprotein model, and compare the marginal distribution of our predictions and ground truth data. In Figure (14.b) we show the distribution of TM-scores for joint sampling of sequence and all-atom structure by the diffusion model. We produce marginal distributions of generated samples Fig.(14.e) and find them to successfully approximate the densities of ground truth data. To test the robustness of joint sampling of structure and sequence, we compute self-consistency TM-Scores[Trippe et al. (2023)]. 54% of our sampled backbones have scTM scores > 0.5 compared to 77.6% of samples from RFDiffusion. We also sample 100 proteins between 50-65 amino acids in 0.21s compared to 11s taken by RFDiffusion on a RTX A6000.

## F.3 Backbone Model

We include metrics on designability in Figure 12.

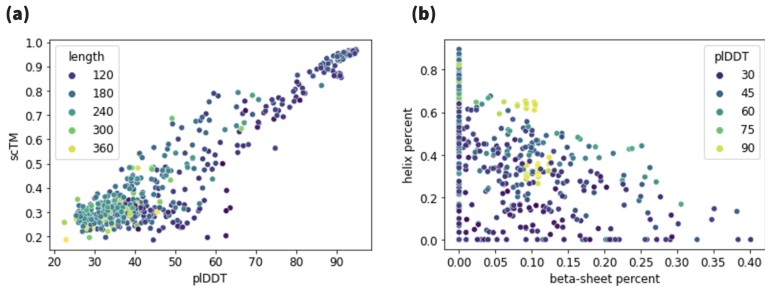

Figure 12: **Designability of Sampled Backbones** . **(a)** To analyze the designability of our sampled backbones we show a plot of scTM vs plDDT. **(b)** To analyze the composition of secondary structures in the samples we show a plot of helix percentage in a sample vs beta-sheet percentage

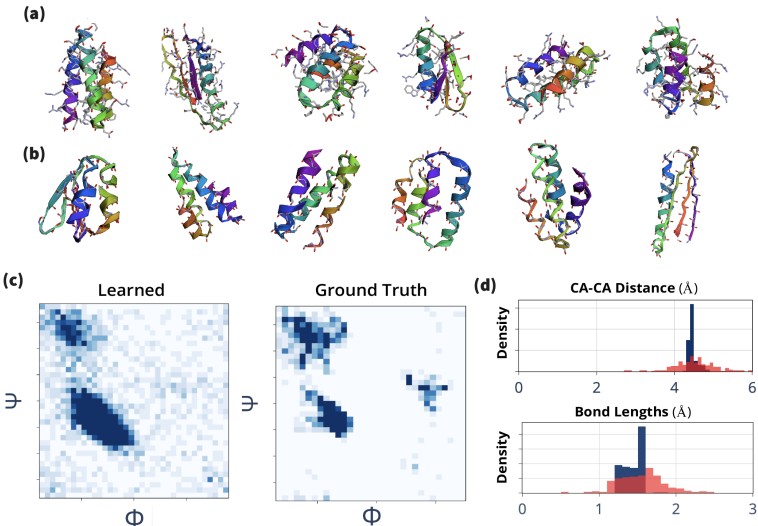

Figure 13: **All-Atom Latent Diffusion.** **(a)** Random samples from an all-atom MiniProtein model. **(b)** Random Samples from MiniProtein backbone model. **(d)** Ramachandran plots of sampled (left) and ground (right) distributions. **(e)** Comparison of C$\alpha$-C$\alpha$ distances and all-atom bond lengths between learned (red) and ground (blue) distributions.

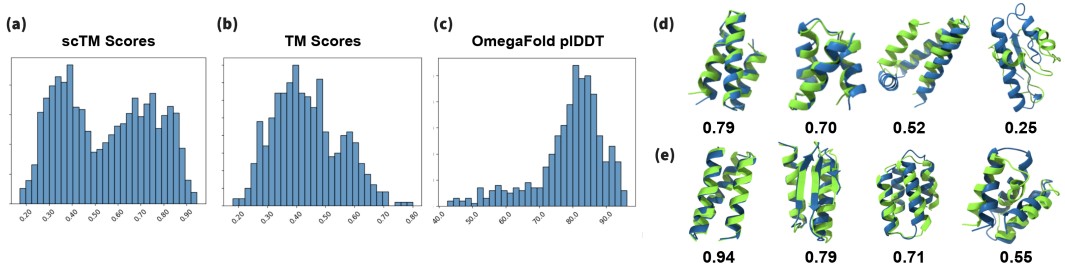

Figure 14: **Self-Consistency Template Matching Scores for Ophiuchus Diffusion**. **(a)** Distribution of scTM scores for 500 sampled backbone. **(b)** Distribution of TM-Scores of jointly sampled backbones and sequences from an all-atom diffusion model and corresponding OmegaFold models. **(c)** Distribution of average plDDT scores for 500 sampled backbones **(d)** TM-Score between sampled backbone (green) and OmegaFold structure (blue) **(e)** TM-Score between sampled backbone (green) and the most confident OmegaFold structure of the sequence recovered from ProteinMPNN (blue)

# G VISUALIZATION OF ALL-ATOM RECONSTRUCTIONS

Figure 15: **Reconstruction of 128-length all-atom proteins**. Model used for visualization reconstructs all-atom protein structures from coarse representations of 120 scalars and 120 3D vectors.

# H VISUALIZATION OF BACKBONE RECONSTRUCTIONS

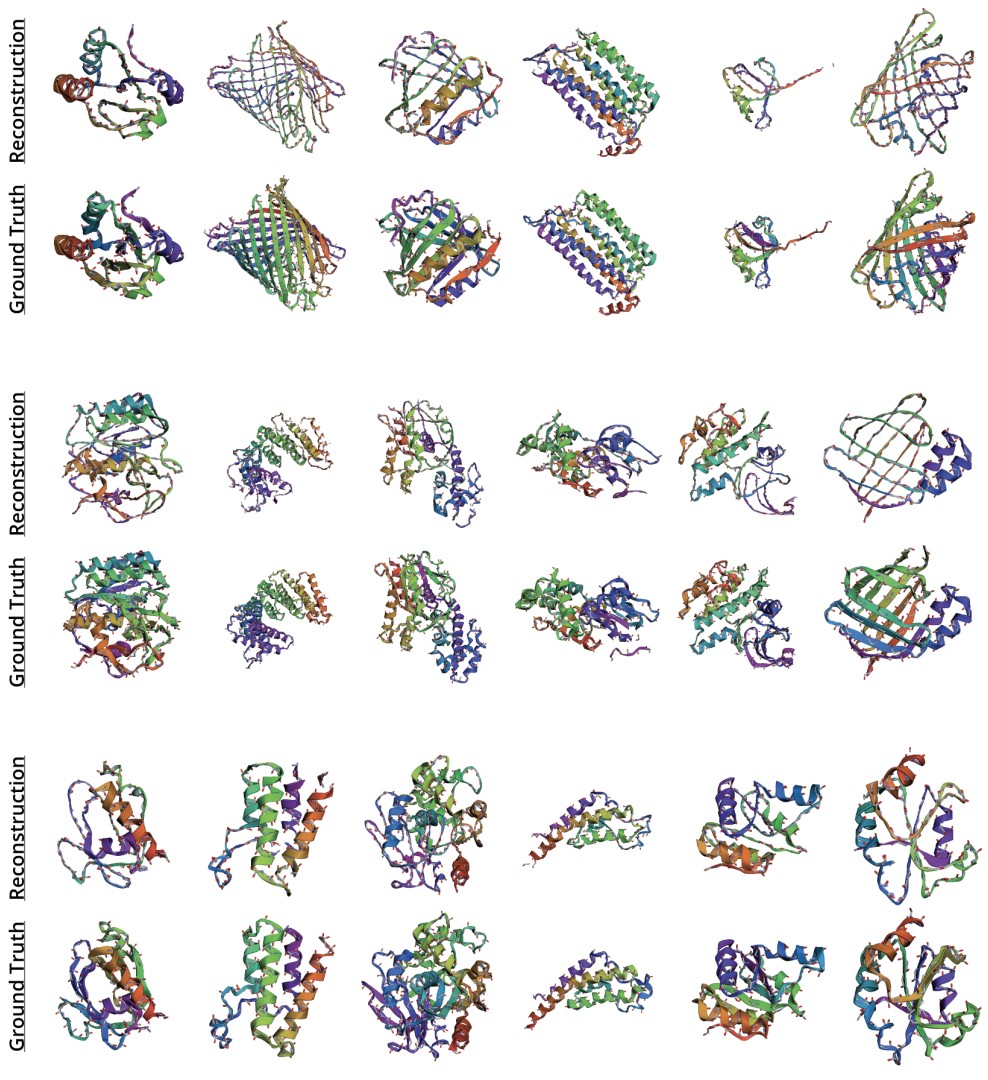

Figure 16: **Reconstruction of 485-length protein backbones**. Model used for visualization reconstructs large backbones from coarse representations of 160 scalars and 160 3D vectors.

# I    VISUALIZATION OF RANDOM BACKBONE SAMPLES

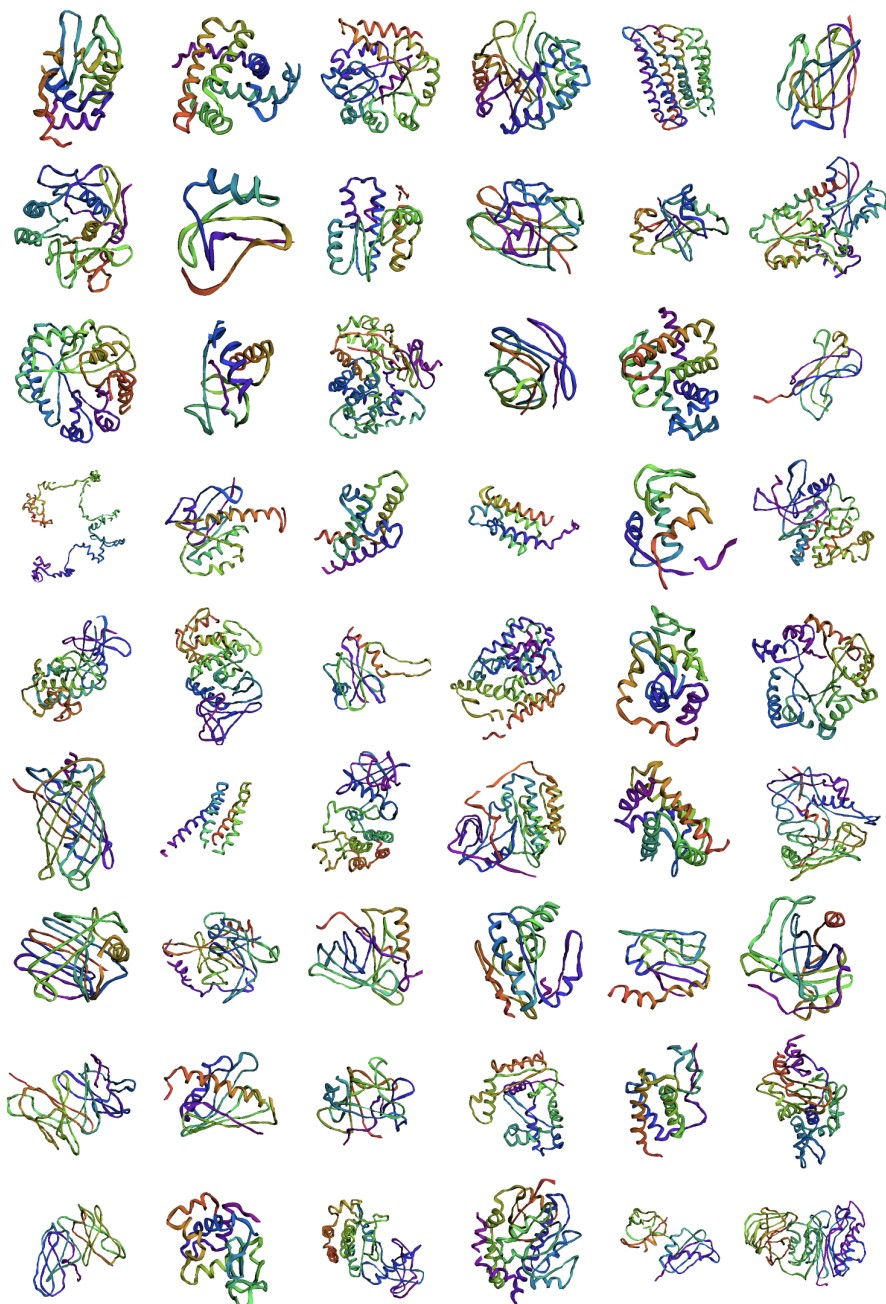

Figure 17: **Random Samples from an Ophiuchus 485-length Backbone Model**. Model used for visualization samples SO(3) representations of 160 scalars and 160 3D vectors. We measure 0.46 seconds to sample a protein backbone in

