# OpenReview forum: "Ophiuchus: Scalable Modeling of Protein Structures through Hierarchical Coarse-graining SO(3)-Equivariant Autoencoders"
_ICLR.cc/2024/Conference — Submitted to ICLR 2024_

### Official Review · Reviewer_eRzw · 2023-10-29

**Soundness:** 3 good
**Presentation:** 2 fair
**Contribution:** 3 good
**Rating:** 3
**Confidence:** 2

**Summary:**

This paper proposes Ophiuchus, a coarse-graining model that is SO(3)-equivariant. This model acts efficiently on the heavy atoms of protein residues. Different from existing methodologies that utilize graph modeling, Ophiuchus prioritizes local convolutional coarsening as a means to represent sequence-motif interactions, which is trained to examine its ability to reconstitute information at various compression rates. The acquired latent space and its application in conformational interpolation for Ophiuchus are studied. Besides, a denoising diffusion probabilistic model is employed to effectively generate decodable latent embeddings of various miniproteins. The results presented in this study show that Ophiuchus possesses the potential to serve as a scalable foundation for the effective modeling and production of proteins.

**Strengths:**

This paper introduces a new autoencoding model for protein sequence-structure representation. This auto-encoder model is examined by comprehensive ablation benchmarks in Table 1 and Table2. Ophiuchus includes a few methods for coarsening, refining, and mixing protein sequence-structure. The generated representations can spawn 3D coordinates directly from features. Moreover, denoising diffusion probabilistic models are leveraged to get the generative model for miniproteins.

**Weaknesses:**

This paper is a little hard to understand. I have tried to make sense of its mechanisms, but there still exist problems for me without its provided codes.
1. In the abstract, the authors say the proposed model focuses on local convolutional coarsening to model sequence-motif interactions in log-linear length complexity. However, I cannot find the complexity analysis in the manuscript.

2. The authors need to examine all the formats of the references that appeared in the paper, whether using () or [] or other formats. Typos: Translation Equivariance of Sequence: One-dimensional Convolutional Neural Networks (CNNs). Colon or dot. In Figure 2, there are two (d).

3. Sometimes, the authors say all-atom protein structures are used, and sometimes, only the heavy atoms are used. It confused me. In Algothrim 1 and Algorithm 2, l=0:2, but in Algorithm 3, l=0:l_{max}. I wonder how to get all atoms' representations from v^{l=0:2}.

4. In Figure 1 (b), why the proposed model uses three building blocks? What do these three building blocks mean? Figure 3 for Protein (PDB ID: 1S9K and PDB ID: 2O93) and Figure 8 for protein (PDB ID: 4JKM and PDB 6LF3) are very similar. I suggest the two pictures be put together. Where are their similarities and differences?


5. The authors say the permutation invariance of particular side-chain atoms is preserved, and a one-dimensional roto-translational equivariant convolutional kernel is designed. Are there any demonstrations to illustrate the invariance and equivariance?

6. In Section 4.1, why choose the contiguous 160-sized protein fragments？ Why train the all-atom generative model for miniproteins instead of large proteins? Why is the model trained in ten epochs?

7. The experiments of conformational interpolation and structure recovery latent diffusion lack comparison methods.

**Questions:**

See above.

---

> ### Author Response · Authors · 2023-11-23
>
> Q1: We added a full analysis of the time complexity of our model in Appendix A.10
>
> Q2: Thanks, we fixed the typos and the format of references.
>
> Q3: In the context of protein modeling, all-atom structures are often interchangeable with "all-heavy atom". Thanks for the note, we made the usage of the term consistent throughout the paper.
>
> Algorithms 1 and 2 are used to encode and decoded the protein into its initial representation which uses degree up to 2. Algorithm 3, however, may be used to produce features of larger degree $l$, through its internal use of tensor square. Because $l_{\textrm{max}}$ is a hyperparameter, the model will build up the geometric representation up to this degree.
>
> Q4: We put the figures together and made a more concise analysis of the two studied interpolations.
>
> Q5: We illustrate the permutation invariance of the side-chain feature parameterization we use. We added a more detailed discussion in Appendix A.1, and incorporated a visual demonstration in Figure 4.
>
> Q6: When choosing our input sequence lengths, we restrict our choices of length to those that will result in perfect alignment of the convolution kernel windows when downsampling down a single representation. For the model settings of our ablation studies, with stride S = 3 and kernel size = 5, the sequence lengths that satisfy this constraint are [1, 5, 17, 53, 160, 485].
>
> We chose ten epochs as we found it to be enough to gather data about training curves and convergence, while still catering to our limited computational resources.
>
> We focused on mini-proteins because our all-atom model performs well at the capacity of our training. For this updated manuscript, we included additional benchmarks on large backbone models that we compare directly to existing architectures.
>
> Q7: To the best of our knowledge, we provide the first demonstration of latent interpolation maintaining consistent chemical validity throughout the trajectory without explicitly enforcing any physical behavior in the latent space. Other methods exist which rely on physics based simulation data for interpolation.
>
> We also added structure recovery comparison in Section 4.1

---

### Official Review · Reviewer_ukWB · 2023-10-31

**Soundness:** 1 poor
**Presentation:** 2 fair
**Contribution:** 1 poor
**Rating:** 3
**Confidence:** 4

**Summary:**

This paper presents a coarse-graining autoencoder for modeling protein structures. The proposed model succeeds in reconstructing the protein structures in multiple resolutions.

**Strengths:**

The problem of modeling protein structures in multiple resolutions is important. The proposed model is well-designed to handle this problem. The presentation is straightforward.

**Weaknesses:**

The biggest problem is that the proposed model is not compared to existing work via experiments. This can be adequate to reject this paper unless the authors can justify why a comparison to other work is not applicable in rebuttal.

Some other problems:
1. Motivations about the model structure design are missing in this paper, including an explanation of why the whole model should contain those substructures and why each structure should be like that.
2. A formal (mathematical) expression of the objective function is missing. The current expression form is too general to give necessary information about how to train the model to authors.
3. The claimed generation power is not evaluated.
3. Some expressions look not academic, e.g. "we introduced a new unsupervised autoencoder" (all autoencoders are unsupervised), "multi-layer hourglass-shaped autoencoder" (all autoencoders are hourglass-shaped, most are multi-layer)

The idea of learning coarse-graining representations for modeling protein structures is interesting and promising. I would encourage the authors to further improve the solidity of this work.

**Questions:**

See weaknesses.

---

> ### Author Response · Authors · 2023-11-23
>
> Thank you for the feedback. Please note the comment above about comparison and baselines.
>
> Q1: We added motivation for the model design at the start of Section 3, as well as at the start description of each component in Sections 3.2-3.4
>
> Q2: We added much more detail on the losses, in Appendix B. In particular, we further elaborate on the chemical losses in Appendix B.2 and include an expression for the final objective function in Appendix B.4
>
> Q3: In Section 4.4, we added comparisons against existing generative models, and added more metrics on diversity and consistency of samples.
>
> Q4: Thank you, we fixed the mistakes.

---

### Official Review · Reviewer_Lcpi · 2023-11-06

**Soundness:** 3 good
**Presentation:** 2 fair
**Contribution:** 3 good
**Rating:** 5
**Confidence:** 3

**Summary:**

The paper proposes a geometric deep learning architecture to process protein structures at different levels of coarsening in a learnable way.
The arcitecuture follows an hourglass design: an encoder downsamples the protein sequence while learning a corsened protein structure and the decoder upsamples the protein sequence while refining the atoms' locations within each residue.
The model employs layers equivariant to 3D rotations and translations.

**Strengths:**

The proposed method is well motivated and, while building on top of existing works, includes some interesting novel ideas (although, I might not be familiar with some related literature).
In particular, the idea of down and up-sample the protein sequence to model the protein structure at different coarsening levels seems novel and particularly useful.

Moreover, I appreciated the extensive quantitative and qualitative studies in the main paper and the appendix, which provided insights into the capabilities of the proposed method.

**Weaknesses:**

The paper doesn't compare empirically the proposed method with other baselines and previous works.
This makes evaluating its benefits challenging.

Moreover, the presentation needs some improvement as the manuscript occasionally misses important details (see Questions below).

**Questions:**

The computational gain of the proposed method is not completely clear.
In Table 1, the lowest downsampling factor shows overall best performance and the models with highest downsampling factors include many more parameters (suggesting they might be more expensive).
Can you quantify and compare the computational gains by using this hourglass design rather than operating at the highest resolutions?
It would also be interesting to compare with a version of the model which doesn't perform any downsampling.


In Sec. 3.1, it is not clear how the $V^{l=2}$ unsigned difference vector is computed and Apx A doesn't include further details about it. Is it obtained by taking the absolute value entry-wise of the difference vector? However, isn't $V^{l=2}$ supposed to be a 5-dimensional vector transforming under the order 2 Wigner D matrix?

The idea that the atoms in standard residues can typically be ordered a priori seems very important but is never explained well, so I think it deserves some additional discussion (in particular, the cases where residues present two-permutations symmetries). However, I am not as familiar with this type of tasks, and this might be well know in the community.

Sec. 3.3 Why is the normalization of the weights ensuring translation equivariance?


Other minor comments and typos:

Sec. 3.1, 4-th line: $P_i^{0, l=0}$  -> $V_i^{0, l=0}$ ?

Sec 3.3: $\hat{P}$ was not previously defined. It could also be worth describing what the HuberLoss is.

---

> ### Author Response · Authors · 2023-11-23
>
> Thank you for the feedback. Please note the comment above about comparison and baselines.
>
> Q1:
> In our ablation studies, it is true that a larger downsampling factor carries more parameters and less recoverability, as is expected in producing larger-sized embeddings and decoding bottlenecked representations. Still, the purpose of our ablation was to estimate the cost of training a model to produce those coarsened embeddings at different resolutions, and quantify the quality of their reconstructions for different embedding sizes.
>
> The main motivation for our novel protein autoencoding architecture is to produce *single* geometric embedding that fully captures the structural features of a protein across multiple resolutions. The inspiration for capturing coarse representations stems from the thermodynamical and evolutionary origin of proteins, and more generally from the need to model building blocks present in structural biology. In our work, we showed that this latent representation of a protein is structured and compact, and can be manipulated while cohesively capturing the structural and chemical features of proteins.
>
> We demonstrate the application of these bottlenecked representations using latent interpolation. To the best of our knowledge, we are the first to explore this form of interpolation for proteins through SO(3)-equivariant features, directly in three-dimensions. The latent interpolation maintains chemical validity throughout its trajectory, as opposed to when the data representation is not downsampled/coarsened and is directly interpolated in the original domain.
>
> The computational gain of our approach starts when the autoencoder model is trained and may be used for downstream tasks. For example, our large backbone model reduces proteins of length 485 into a single geometric tensor representation of 160 scalars and 160 3D vectors, reducing the target size for denoising diffusion. This directly reflects in sampling times orders of magnitude lower than existing models, while performing comparably.
>
> As a final example of the usefulness of our structured latent space, we note that the coarsened embeddings capture sequence labels, all-atom positions, and backbone structure, and our generative model is able to produce all of those simultaneously while diffusing on single representation of features. This contrasts to existing models, which often are required to perform separate diffusion processes on multiple data domains.
>
> Q2: Please refer to the updated manuscript. We added much more detail on the encoding of $\mathbf V^{l=2}$ in Section 3.1, and added diagrams (Figure 4) and further discussion in Appendix A.1.
>
> Q3: In Section 3.1, and Appendices A.1 and A.2, we added details and discussion about the permutation symmetries. In Appendix A.1, Figure 5, we also added a visual demonstration that shows how our proposed feature representations handle the problems that might arise due to the symmetries of residues.
>
> Q4: We added Appendix A.5 where we derive the conditions for coarsening coordinates while satisfying rotation and translation equivariance.
>
> Q5: Thanks, we fixed the minor issues.
>
> Q6: We added a brief description of the HuberLoss in Appendix B.1.

---

### Author Response · Authors · 2023-11-23
**Comparison to other models as baselines**

We appreciate the feedback, and we are happy to provide additional clarification and conduct additional experiments to further demonstrate the utility of our work.

First, we added Section 4.1, where we compare our autoencoder model to the closest architecture that does the protein reconstruction task in three-dimensions, a recently proposed modification of EGNN. We provide comparisons for different settings of downsampling and show the significant advantage our model has over the more commonly used EGNN architecture.

In Section 4.4, we also scaled our model and compared it to state-of-the-art architectures for sampling protein structures. We measured commonly used metrics to evaluate the generated samples. We found our model to produce comparable results in terms of quality, while enabling orders of magnitudes faster sampling speeds.

---

### Meta-Review · Area_Chair_8VgQ · 2023-12-11

**Metareview:**

The paper introduces a novel SO(3)-equivariant autoencoder for protein structure modelling, employing graph-based methods to learn coarsened and refined representations. The core contribution lies in the novel model, an extensive ablation study of the model and a novel. The reviewers acknowledged the method's originality and potential and the extensive quantitative and qualitative studies. Despite these strengths, they uniformly identified significant shortcomings in the paper. Key among these was the need for robust empirical comparisons with existing models, insufficient clarity in the exposition of the method, and a comprehensive evaluation that falls short, particularly in assessing the model's generative power and proficiency in handling permutation invariance. The authors addressed many of these concerns in their revisions, which improved the overall quality of the paper. However, some reviewers remained unsatisfied with the responses, indicating that while improvements were made, some issues may not have been fully resolved.

**Justification For Why Not Higher Score:**

The unanimous critical feedback from the reviewers, with none recommending acceptance, supports the recommendation of rejecting the paper.

**Justification For Why Not Lower Score:**

N/A

---

### Decision · Program_Chairs · 2024-01-16

Reject